# High-precision and linear weight updates by subnanosecond pulses in ferroelectric tunnel junction for neuro-inspired computing

Zhen Luo[1,3], Zijian Wang[1,3], Zeyu Guan[1,3], Chao Ma[1], Letian Zhao[1], Chuanchuan Liu[1], Haoyang Sun[1], He Wang[1], Yue Lin [1], Xi Jin[1], Yuewei Yin [1✉] & Xiaoguang Li [1,2✉]

The rapid development of neuro-inspired computing demands synaptic devices with ultrafast speed, low power consumption, and multiple non-volatile states, among other features. Here, a high-performance synaptic device is designed and established based on a Ag/ PbZr$_{0.52}$Ti$_{0.48}$O$_3$ (PZT, (111)-oriented)/Nb:SrTiO$_3$ ferroelectric tunnel junction (FTJ). The advantages of (111)-oriented PZT (~1.2 nm) include its multiple ferroelectric switching dynamics, ultrafine ferroelectric domains, and small coercive voltage. The FTJ shows high-precision (256 states, 8 bits), reproducible (cycle-to-cycle variation, ~2.06%), linear (non-linearity <1) and symmetric weight updates, with a good endurance of >10$^9$ cycles and an ultralow write energy consumption. In particular, manipulations among 150 states are realized under subnanosecond (~630 ps) pulse voltages ≤5 V, and the fastest resistance switching at 300 ps for the FTJs is achieved by voltages <13 V. Based on the experimental performance, the convolutional neural network simulation achieves a high online learning accuracy of ~94.7% for recognizing fashion product images, close to the calculated result of ~95.6% by floating-point-based convolutional neural network software. Interestingly, the FTJ-based neural network is very robust to input image noise, showing potential for practical applications. This work represents an important improvement in FTJs towards building neuro-inspired computing systems.

[1] Hefei National Laboratory for Physical Sciences at the Microscale, Department of Physics, and CAS Key Laboratory of Strongly-Coupled Quantum Matter Physics, University of Science and Technology of China, Hefei, China. [2] Collaborative Innovation Center of Advanced Microstructures, Nanjing University, Nanjing, China. [3] These authors contributed equally: Zhen Luo, Zijian Wang, Zeyu Guan. ✉email: yyw@ustc.edu.cn; lixg@ustc.edu.cn

Neuro-inspired computing shows promise for application in accomplishing data-centric cognitive tasks, including real-time image recognition and decision making, which are very important for edge computing at Internet of Things (IoT) terminals, such as traffic sign recognition and speed control for intelligent vehicles[1–3]. However, the efficiency in terms of both energy and time has become problematic for executing the corresponding computations on the conventional von Neumann computing system because the data have to be calculated by a processer, stored in a memory, and transferred between the memory and the processer[2,3]. It is commonly believed that to realize an efficient neuro-inspired computing system, it is necessary to develop high-performance synaptic devices that are based on memristors and capable of emulating the weight updates of biological synapses[4,5].

To date, various types of memristor-based synapses have been reported, such as phase-change memristors, magnetic tunnel junction (MTJ) memristors, and resistive memristors. However, a phase-change or MTJ memristor needs to be operated with a high-density current and suffers from high energy consumption[6–13], while a resistive memristor, based on defects, can show undesirable variations[12,13]. Therefore, artificial synapses meeting the desired specifications (summarized in Supplementary S1)[14–16] are still scarce, which limits the performance of corresponding neuro-inspired neural network computing systems. For example, the basic tasks of recognizing handwritten digits in the Modified National Institute of Standards and Technology (MNIST) database and more complicated fashion product images in the Fashion-MNIST (F-MNIST) database have been widely used to test the capability of neural network computing systems[17,18]. Memristor-based neural networks in simulations and experiments typically show degraded recognition fidelity because of the imperfection of conductance manipulation in these devices[14]. The situation is even more serious when noisy images have to be classified, which is especially important in practical applications, such as speed sign recognition in bad weather (e.g., rain or snow). Thus, it is necessary to develop a high-performance memristor that meets the desired performance criteria.

As a recently developed memristor strategy, the ferroelectric tunnel junction (FTJ) is in principle a promising candidate for building high-performance artificial synapses. This is because the FTJ stores data non-volatilely and intrinsically as ferroelectric polarization states in its ultrathin ferroelectric barrier. Thus, the conductance can be continuously manipulated by ferroelectric domain switching with less variation and low current densities[19–23]. Recently, $Pt/BaTiO_3$ (001)/$Nb:SrTiO_3$ (NSTO)-based FTJs with 200 states in a conductance dynamic range of 10× have been reported with an operation speed of 50 ns and an endurance of >$1.1 \times 10^4$ cycles[24]. However, the reported FTJ synapses still do not meet the target specifications listed in Supplementary S1, and the following aspects need to be improved.

First, more conductance states in a sizable range are required for an FTJ synapse to realize precise weight updates[14]. It has been proposed that to train a relatively large neural network, each synaptic device of the network should have a precision of at least 8 equivalent bits[25]. Based on the principle of the FTJ, multiple conductance states are related to multiple ferroelectric domains. The successive switching process of multiple domains will lead to a gradual manipulation of conductance[26]. Thus, shrinking the lateral size of the ferroelectric domain will be beneficial for achieving more conductance states, and two strategies could be utilized. (i) Reducing the thickness of the ferroelectric barrier is generally conducive to forming ferroelectric polydomains due to the higher depolarization electric field[27]. (ii) The crystalline orientation can also influence the domain structure and its switching dynamics. For example, it has been reported that the (111)-oriented ferroelectric titanate film is promising for

constructing polymorphic nanodomains[28] and multistep switching processes[29,30]. Therefore, compared with typically reported (001)-oriented FTJs, FTJs with ultrathin (111)-oriented ferroelectric titanate films would be capable of realizing more conductance states but have not yet been reported.

Second, an ultrafast operating speed under an affordable voltage is important for building a high-performance neuro-inspired computing system. In particular, considering that synaptic devices are updated frequently during online training, a subnanosecond operating speed that is comparable to that of a central processing unit (CPU) would be beneficial for constructing a high-speed neuromorphic computing system[14]. However, most reported artificial synapses have been manipulated using voltages with pulse durations ≥10 ns[14,31]. Very recently, we produced a $Ag/BaTiO_3$ (001)/NSTO-based FTJ memristor with a subnanosecond operating speed (600 ps) and a low current density ($4 \times 10^3 A/cm^2$)[32], but the operation voltage was above 10 V, limiting its practical applications. To solve this problem, decreasing the ferroelectric film thickness and especially choosing a material with a lower coercive field to reduce the ferroelectric coercive voltage may be viable. Ferroelectric materials near morphotropic phase boundaries (MPBs), such as $PbZr_{0.52}Ti_{0.48}O_3$ (PZT), typically have low coercive fields[33], and (111)-oriented PZT has an even smaller coercive field than (001)-oriented PZT[34]. Thus, in addition to the potential for achieving more conductance states, FTJs with ultrathin (111)-oriented PZT barriers have the potential to yield subnanosecond switching speeds at low voltages (≤5 V).

In this work, according to the above discussion, we designed and constructed a high-performance FTJ synapse based on a Ag/PZT ((111)-oriented, ~1.2 nm)/NSTO (Nb: 0.7 wt%) heterostructure. By selecting the ultrathin (111)-oriented PZT that is near the MPB as the ferroelectric barrier, the analog manipulation of 150 conductance states can be achieved by applying voltage pulses with a duration ($t_d$) as fast as 630 ps and a low $V_p \le 5$ V. Resistance switching speed of 300 ps is achieved by voltages <13 V, which is the fastest switching speed among reported FTJs[32,35,36]. Notably, the target specifications for the artificial synapse listed in Table S1 are achieved, including numerous states (256), a sufficient conductance dynamic range (~100×), high switching endurance ($10^9$), low energy consumption per programming step (~5.3 fJ for a 50-nm-diameter FTJ), low cycle-to-cycle variation (2.06%) and linear (nonlinearity <1) conductance manipulation. Based on the performance of the experimental device, the simulated convolutional neural network (CNN) can achieve a high online learning accuracy ~94.7% for recognizing F-MNIST images, which is close to the result of ~95.6% obtained by floating-point-based CNN software. High recognition accuracy of >90% can still be realized for recognizing noisy F-MNIST images with a certain salt & pepper or Gaussian noise, suggesting its practical potential for neuromorphic computing.

## Results

**Structural and ferroelectric characterizations**. The device structure of the FTJ with a (111)-oriented PZT barrier is schematically illustrated in Fig. 1a. The (111)-oriented PZT was epitaxially grown on the (111)-oriented NSTO substrate (see Methods for details). The voltage was applied to the top silver electrode with a diameter of ~100 μm. The NSTO substrate was always grounded during the application of voltage. Figure 1b depicts the high-angle annular dark-field scanning transmission electron microscopy (HAADF-STEM) images selected from 4 different areas viewed along the [01$\bar{1}$] direction. It is shown that the thickness of the ferroelectric barrier is ~1.2 nm. As indicated by the ferroelectric atomic displacements in the magnified images shown in the insets of Fig. 1b, different atomic polarization

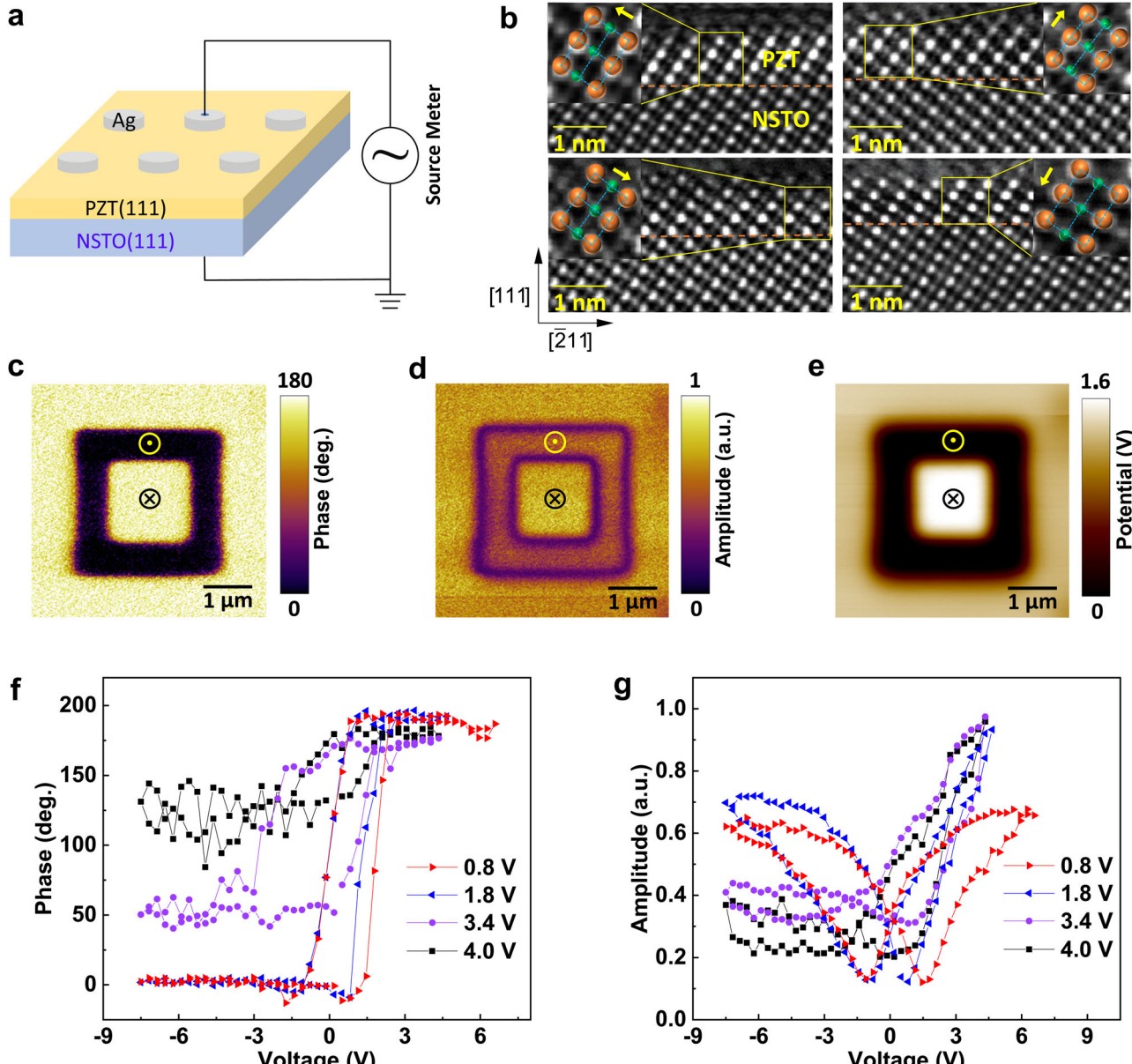

**Fig. 1 Structural and ferroelectric properties. a** Schematic illustration of Ag/PZT/NSTO FTJ devices. **b** Cross-sectional HAADF-STEM images of the PZT/ NSTO heterostructure at four different areas, with the insets showing the ferroelectric atomic displacements in the PZT. The orange and green spheres denote Pb and Zr/Ti ions, respectively. The arrows in the insets indicate the polarization directions. **c** PFM phase, **d** PFM amplitude, and **e** SKPM surface potential images recorded after writing an area of $3 \times 3$ μm$^2$ with $-6$ V and the central area of $1.5 \times 1.5$ μm$^2$ with $+5$ V on the (111)-oriented PZT (1.2 nm)/ NSTO. **f** PFM phase and **g** amplitude loops collected with various AC voltages.

directions are observed for these domains in the (111)-oriented PZT ferroelectric film because of the preferred [100], [010] or [001] ferroelectric polarizations of PZT in principle[37].

The ferroelectric properties and domain structures were probed by piezo-response force microscopy (PFM)[38] and scanning Kelvin probe microscopy (SKPM)[39]. Fig 1c, d and e show the PFM phase, amplitude and associated SKPM potential images, respectively. These mappings were carefully tested after the box-in-box ferroelectric domain patterning by reversed tip biases of $-6$ V and $+5$ V. As the PFM phase mapping shown in Fig. 1c, the phase contrast between poled-up and poled-down regions reaches 180 degree. And the domain boundaries can be clearly seen in the PFM amplitude image. These evidence the reversible switching of ferroelectric domains in the PZT ultrathin film. The SKPM mapping in Fig. 1e shows that the

surface potential in the poled-up region is lower than that of the poled-down region, because the polarization charges are over-compensated by screening charges injected from the tips, consistent with previous SKPM results on BiFeO$_3$ and PZT ferroelectric films[39–41]. As shown in Fig. 1f and g, the PFM phase and amplitude hysteresis loops were characterized with different AC voltages. The deformed loops with increasing AC voltage further evidence the robust ferroelectricity of the ultrathin PZT film[36,42].

The evolution of the ferroelectric polydomains for PZT films with reducing thickness (d) was verified by PFM, as shown in Fig. 2a. Consistent with earlier reports[43], the domain sizes decrease with decreasing ferroelectric film thickness, and the spontaneous ferroelectric domain size for $d = 1.2$ nm PZT thin film can be estimated to be ~12 nm. It should be noted that the

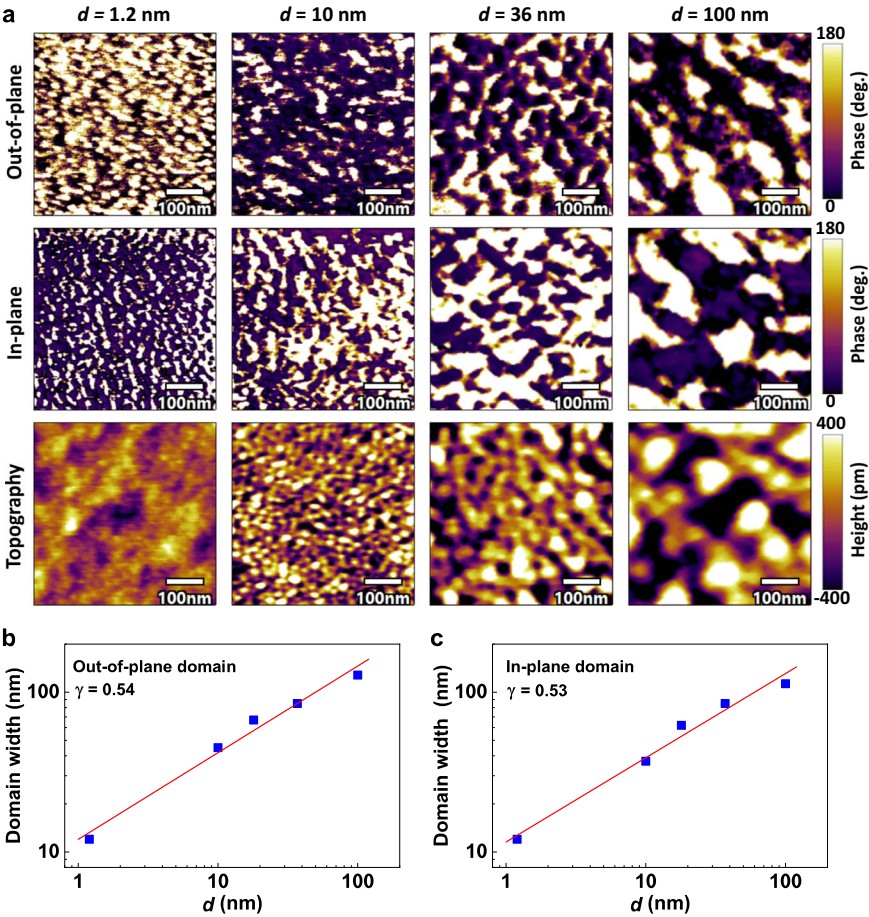

**Fig. 2 Ferroelectric domain structures of (111)-oriented PZT films with different thicknesses. a** PFM out-of-plane and in-plane phases and topographies of PZT films with different thicknesses. **b, c** Domain width *versus* film thickness (*d*) for out-of-plane and in-plane, respectively. The solid lines are the fitting results by a power law.

real domain size may be even smaller because the measurement resolution is limited by the diameter of the conductive PFM tips[44]. Interestingly, as shown in Fig. 2b, c, the domain width (*W*) as a function of PZT thickness can be fitted by a power law:

$$W = Ad^{\gamma} \tag{1}$$

with a scaling exponent $\gamma = 0.54$ and 0.53 for the out-of-plane and in-plane domains, respectively. This follows the famous Landau-Lifshitz-Kittel scaling law[43] with $\gamma = 0.5$. These results demonstrate the important role of reducing film thickness in obtaining ultrafine ferroelectric domains which are beneficial for realizing multilevel resistances in FTJ devices.

**Two-step ferroelectric domain switching dynamics**. Due to the multidomain structure and complex domain reversal dynamics in (111)-oriented PZT[29], the corresponding FTJs will show continuous resistance switching characteristics as a result. The hysteresis *I-V* curve in Supplementary Fig. S1a indicates the obvious memristive behavior in the FTJ. To realize the relationship between ferroelectric reversal and resistance switching, it is necessary to analyse the ferroelectricity-affected band structures. Thus, the *I-V* curves at different temperatures from 150 to 270 K were measured at ON and OFF states, and the thermally assisted tunneling model was used to fit the data, as shown in Supplementary Fig. S1b–e. When the ferroelectric polarization points to the NSTO, a lower Schottky barrier height with a narrower depleted region is obtained in the ON state, consistent with the ferroelectric field effect mechanism in a metal/ferroelectric/ semiconductor FTJ[36].

Because of the ferroelectricity-affected band structure of the FTJ, the resistive switching can be directly linked with ferroelectric polarization switching. Thus, in turn, the ferroelectric switching dynamics can be investigated by measuring the evolution of resistance under voltage pulses[35]. For the measurement, the FTJ was first set to the lowest resistance state (ON state, downward polarization) by a voltage pulse $V_{set} = 2$ V (duration $t_d = 1$ μs), and then pulsed voltages with varying amplitudes ($V_p$) and durations ($t_d$) were applied before the resistance measurements by a read voltage $V_{read}$ of 0.05 V, as schematically shown in Fig. 3a. The relationship between the resistance of the FTJs and the pulse duration at different amplitudes is shown in Fig. 3b. The resistance evolves slowly at a small $|V_p|$ value of 1.5 V but increases sharply with increasing $|V_p|$. Interestingly, there are two switching steps in the resistance dynamic curves for $|V_p| \leq 2.8$ V, as indicated by the kinks denoted by the arrows in Fig. 3b. With increasing $|V_p|$, the kink appears at a smaller $t_d$ with a higher resistance and finally disappears when applying $|V_p| \geq 3.2$ V. Importantly, a significant resistance switching of 500% from the ON state can be realized by a 630 ps ultrafast pulse voltage as small as −4 V. This pulse voltage is much smaller than that in a previous report (~10 V)[32] because of the low coercive voltage in the ultrathin (111)-oriented PZT ferroelectric layer. This resistance evolution process is similar to that of the ferroelectric polarization dynamics reported in (111)-oriented PbZr$_{0.2}$Ti$_{0.8}$O$_3$ (80 nm thickness) films, which was attributed to the multiple ferroelectric switching processes[29,30].

The ferroelectric domain switching dynamics can be obtained directly from the resistance switching dynamics by considering

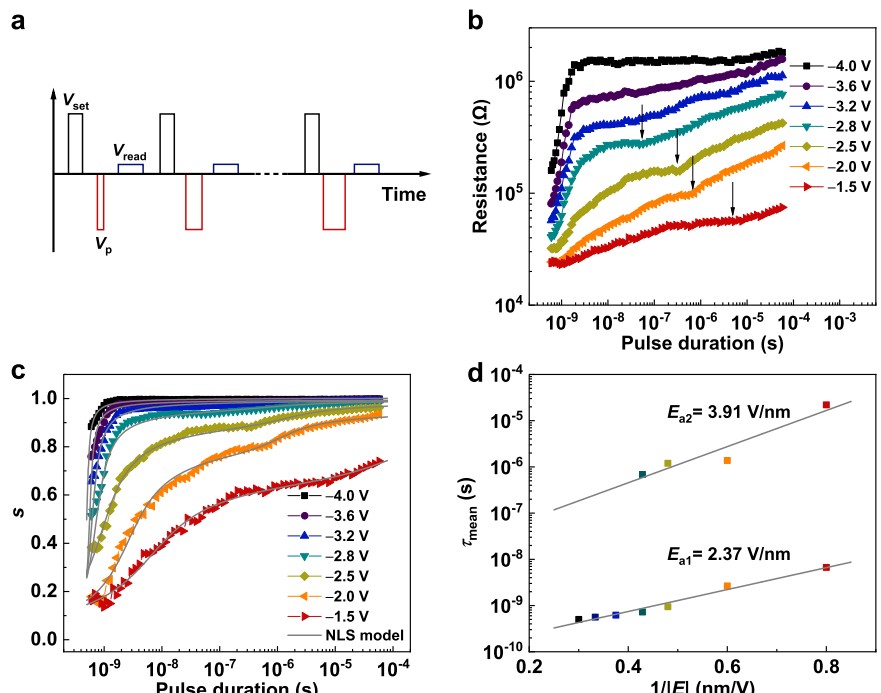

**Fig. 3 Ferroelectric domain switching dynamics for the (111)-oriented PZT in FTJs. a** Schematic illustration of the applied voltage pulse sequence. **b** Resistance measured at 0.05 V *versus* pulse duration. The kinks are denoted by arrows. **c** Relative area fraction of the ferroelectric poled-up domain *versus* pulse duration. The solid curves are the results of fitting by the nucleation-limited-switching (NLS) model. **d** Evolution of the mean switching time ($\tau_{mean}$) as a function of the inverse of electric field ($1/|E|$). The solid lines are the fitting results by Merz's law.

the FTJ as a parallel circuit of upward and downward polarization domains[35]. The relationship between the area fraction $s$ for upward polarization domains and the junction resistance can be expressed as $1/R = (1 - s)/R_L + s/R_H$ [35,45]. Fig 3c shows that the pulse duration dependence of $s$ at different $V_p$ has a plateau, suggesting the multistep ferroelectric switching process. According to Martin et al.'s report[30], in (111)-oriented PZT under an external electric field, some fraction of domains undergoes a one-step 180° switching process, while the other fraction of domains undergoes a two-step 90° switching process. The 90° switching process occurs at the plateau where the out-of-plane component of polarization is invariable. For FTJs with a (001)-oriented PZT barrier, there is no plateau during switching (see Supplementary Fig. S2), and only a one-step switching process occurs. The two-step switching dynamics in FTJs with (111)-oriented PZT will be beneficial for achieving multiple and stable intermediate ferroelectric domain states. In addition, to describe the switching dynamics for (111)-oriented PZT, the nucleation-limited-switching (NLS) model[35,46] containing two steps (see Methods for details) was used to fit the results in Fig. 3c, and the mean switching time $\tau_{mean}$ was extracted accordingly. The relationships between $\tau_{mean}$ and the inverse of the applied electric field $1/|E|$ for the two-step switching processes are shown in Fig. 3d. The two activation electric fields of 2.37 V/nm and 3.91 V/nm, corresponding to the first and second switching steps, respectively, were obtained according to Merz's law $\tau \propto \exp(-E_a/E)$ [23].

**FTJ-based analog memristors**. To further reveal the characteristics of the memristive behavior, the loops of resistance ($V_{read} = 0.05$ V) *versus* $V_p$ were investigated. The representative results measured at $t_d = 10$ ns and 630 ps are shown in Fig. 4a, b, respectively. The resistance can be manipulated continuously to various intermediate resistances by varying the negative maximum voltage $V_p^{max-}$. As shown in Fig. 4c, d, the stable resistance switchings among different states can be established by

changing the applied voltage. A high ON/OFF ratio of ~200 can be realized by applying 3.6 V/−4.0 V at $t_d = 10$ ns or 8.0 V/−8.5 V at $t_d = 630$ ps. Even voltages as low as 3.6 V/−3.6 V, much lower than previously reported voltages[32], are sufficient to achieve a distinguishable resistance switching repeatedly at such a high switching speed of ~630 ps. The subnanosecond switching speed under a small operating voltage is conducive to saving a significant amount of time and energy in training a neural network computing system because the synaptic device conductance would be updated frequently during online training[47].

For our FTJs, the lower operation voltages than those of previous report on BaTiO₃-based FTJ[32] should be attributed to the relatively lower coercive field for the (111)-orientated PZT near MPB[34] as well as its ultrathin thickness. Supplementary Fig. S3 shows the resistance *vs.* $V_p$ loops of FTJs with different PZT thicknesses $d$ from 6.0 nm to 1.2 nm under voltage pulses of $t_d = 10$ ns and 630 ps. It is revealed that the coercive voltages ($V_c$) of our FTJs decrease with reducing the thickness of PZT, and their relationship can be described by the Kay-Dunn law $V_c \propto d^{1/3}$ for ferroelectric thin films[34]. Interestingly, it is noted that with decreasing ferroelectric film thickness, the FTJ resistance decreases exponentially, indicating the tunneling effect in FTJs[48]. In addition, the utilization of metal electrode with low work function, such as Ag ~4.26 eV, is also beneficial to reduce the operation voltage. To investigate the effect of electrode on the FTJ performance, FTJ devices with other electrodes including Cu (CMOS-compatible[49]) and Pt were studied, as shown in Supplementary Fig. S4 and Table S2. With increasing work functions of electrodes from Ag to Pt, the ON/OFF ratio increases, but the operation voltages increase obviously. Because of the low operation voltage, the Ag electrode FTJ shows more robust switching endurance, as shown in Supplementary Fig. S4. In addition, it is worth mentioning that although Ag migration may cause resistance switching in diffusive memristors[50,51] or conductive bridge memories[52], the experimental results

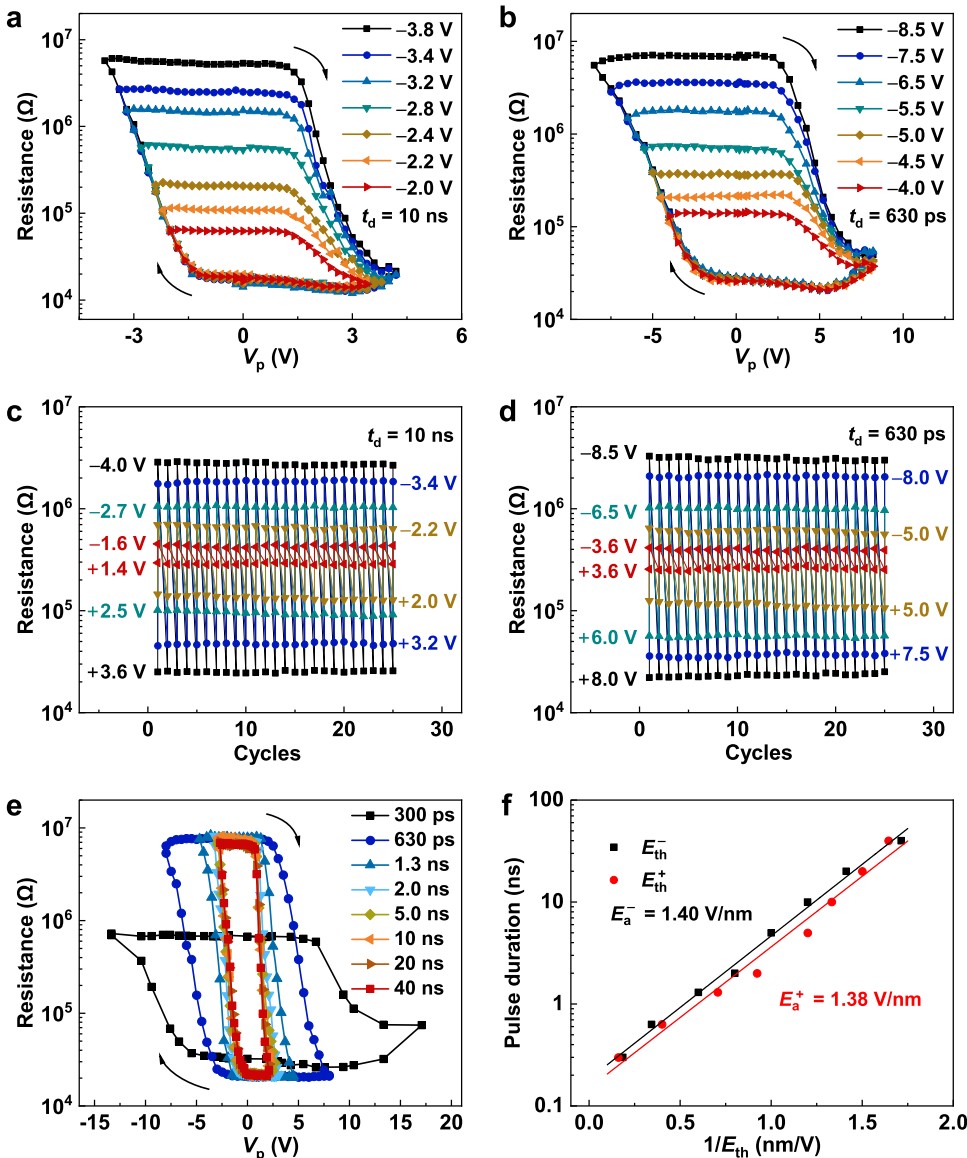

**Fig. 4 Ultrafast memristive switching of the FTJ with a (111)-oriented PZT barrier. a**, **b** Resistance measured at 0.05 V as a function of $V_p$ with a pulse duration of $t_d = 10$ ns and $t_d = 630$ ps. **c, d** Resistance switchings among different resistance states by applying different $V_p$ with $t_d = 10$ ns and $t_d = 630$ ps. **e** Resistances as a function of $V_p$ with various $t_d$ values from 300 ps to 40 ns. **f** Pulse duration *versus* inverse of threshold field $1/|E_{th}|$. The solid lines are the results of fitting by Mertz's law. The arrows in **a**, **b** and **e** indicate the voltage sweeping direction.

confirm that the resistance switching of our FTJ is caused by ferroelectric polarization switching rather than the formation and rupture of Ag conductive bridge (see detailed descriptions in the Supplementary S6).

With the decreased operation voltage, the current density of FTJ can be reduced further to ~$1.3 \times 10^3$ A/cm$^2$ compared with earlier FTJs[32], and this is much lower than the values for phase-change memristors and MTJ memristors[6,9]. As a result, the energy consumptions of ~440 pJ per positive operating pulse and ~520 pJ per negative operating pulse were obtained in the FTJ with diameter of 100 μm, as shown in Supplementary S7. To further decrease the energy consumption, nanoscale FTJ devices with a top electrode diameter of ~50 nm were prepared, with an energy consumption as low as 5.3 fJ/bit, as shown in Supplementary S8.

Fig 4e shows that the higher the applied amplitude of $V_p$ is, the quicker the resistance switching is achieved, consistent with the dynamics shown in Fig. 3. In particular, an operation

speed of 300 ps (the fastest resistance switching speed for FTJs) was achieved by voltages <13 V, as the $R–V_p$ loops and multi resistance state switchings shown in Fig. 4e and Supplementary Fig. S8. The threshold electric fields $E_{th}$ (defined as the electric field where the resistance is 50% higher (or lower) than that at the lowest (or highest) resistance state) of resistance switching at different $t_d$ were extracted and plotted, as shown in Fig. 4f fitted by Merz's law[53]. The activation field $E_a$ are 1.40 V/nm for negative voltages and 1.38 V/nm for positive voltages, respectively.

Retention and endurance characteristics are two critical parameters for electronic synapses. Fig 5a shows the retention up to $10^4$ s for representative 4-bit resistance states. Fig 5b shows the good endurance up to $10^9$ cycles for the FTJ with a (111)-oriented PZT barrier, meeting the endurance requirement for neural network training (see Table S1)[14]. Although during each iteration of online learning, the device conductance only needs to be modified by an incremental amount and not every synapse is

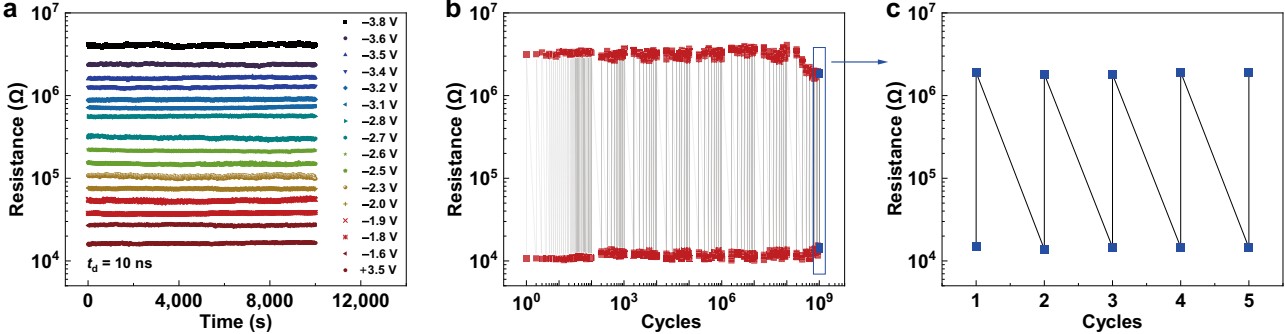

**Fig. 5 Retention and endurance measurements of the FTJ with a (111)-oriented PZT barrier. a** Retention time of representative 4-bit resistance states. **b** Endurance by applying $+2\,V/-3\,V$ with pulses of $t_d = 100$ ns. **c** 5 resistance switching cycles after the $10^9$ cycles.

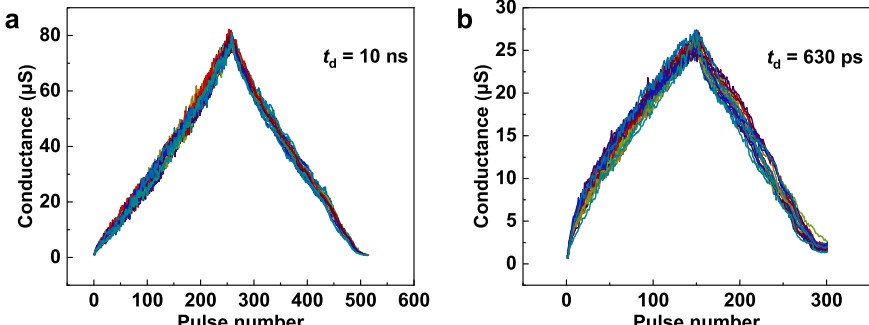

**Fig. 6 Artificial synapse analog based on the FTJ with a (111)-oriented PZT. a**, **b** Potentiation and depression processes measured for 20 times from one FTJ sample with pulse durations of $t_d = 10$ ns and $t_d = 630$ ps, respectively.

updated, synaptic devices with an endurance of up to $10^9$ cycles are highly desired for more complex tasks, such as reinforcement learning[16,54]. In addition, it is worth mentioning that in Fig. 5b, after applying $10^9$ cycles of voltage pulses, another 5 cycles of resistance switching were measured as shown in Fig. 5c. It can be seen that the FTJ was not damaged but fatigued with the ON/OFF ratio decreasing from ~300 to ~140. The high endurance of our FTJ should be related to the high quality of epitaxial ferroelectric film with less grain boundaries (see Supplementary Fig. S10)[55].

**Subnanosecond weight manipulation of the FTJ synapse with pulsed voltages ≤5 V.** Based on the gradual manipulation of the conductance, the FTJ memristor can be harnessed to construct an artificial synapse. As shown in Fig. 6a, both potentiation and depression with 256 conductance states (the highest number of states among reported FTJs) in a dynamic range of 100× were demonstrated by applying a series of 10 ns pulse voltages with incremental amplitudes (from 1.35 V to 2 V for potentiation, and from −1.4 V to −3.5 V for depression), and the operation speed is comparable to that of DRAM[56]. Here, the variable voltage scheme was used to tune the conductance, because it is beneficial for improving the linearity of conductance manipulation (non-linearity ~0.77 for potentiation, ~−0.94 for depression), as discussed in Supplementary S11. In addition, the measurements were repeated by 20 times for one FTJ sample, and due to the intrinsic stability of ferroelectricity, the FTJ shows a low cycle-to-cycle variation of ~2.06% (see Supplementary Fig. S11 for the mean value and standard deviation of each conductance state). When the pulse is as fast as 630 ps (close to CPU speed), as shown in Fig. 6b, 150 conductance states with a cycle-to-cycle variation of 3.65% can be demonstrated by applying a series of operation pulsed voltages ≤5 V (from 3.5 V to 4.5 V for potentiation, and from −3.2 V to −5 V for depression). The subnanosecond

operation speed and low operation voltage of ≤5 V show great advantages in saving training time, which is significant for developing high-speed neuromorphic computing systems.

## Discussion

**Convolutional neural network simulations based on the experimental performance of FTJ synapse.** To further illustrate the capability of the FTJ with a (111)-oriented PZT barrier as a high-performance synapse, the neural network simulations were carried out based on the experimentally obtained behaviors including 256 states with 10 ns pulse duration or 150 states with 630 ps pulse duration[57]. As shown in Fig. 7a, the convolutional neural network ResNet-18 was established to recognize fashion product images in F-MNIST dataset[17,58]. The device behavioral models for CNN simulations were constructed based on the experimental conductance manipulations in Fig. 6a, b with considering the experimental cycle-to-cycle variation and nonlinearity summarized in Table S3. The more detailed simulation processes are discussed in Methods and Supplementary S12. The simulation results are shown in Fig. 7b. The CNN simulation based on 256 states shows a high recognition accuracy of 94.7% for F-MNIST, which is close to that ~95.6% achieved by floating-point-based software, demonstrating the excellent performance of our FTJ synapses. When the simulation is performed based on 150 states, the recognition accuracy decreases, but is still higher than 90.0%. In addition, the CNN simulations on recognizing handwritten digits in the MNIST dataset were also carried out (see Supplementary S12). The high accuracies of 99.7%, 99.5% and 99.1% for the simple MNIST dataset were achieved based on floating point, 256 and 150 states, respectively. These simulation results further evidence the advantages of FTJs for neural network.

It is worth mentioning that instead of recognizing clear images in the F-MNIST or MNIST database, a practical neuromorphic

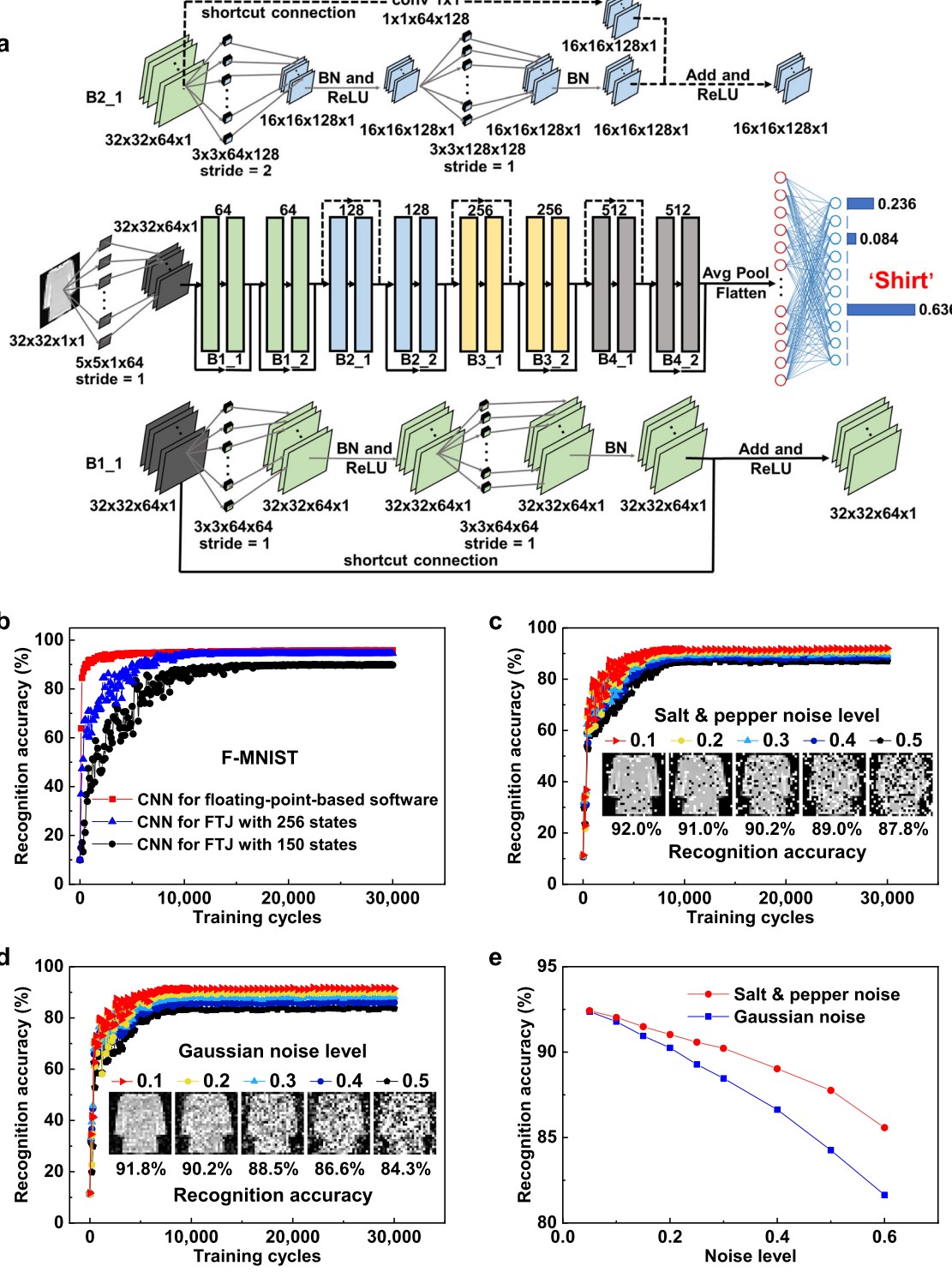

**Fig. 7 Neural network simulation. a** Schematic diagram of the ResNet-18 neural network. **b** Simulation results on learning F-MNIST images based on the experimental results of 256 (in Fig. 6a) and 150 (in Fig. 6b) conductance states as well as floating-point-based software. **c**, **d** Training results on F-MNIST images with different levels of salt & pepper noise and Gaussian noise, respectively. **e** Recognition accuracy of F-MNIST images with different levels of salt & pepper noise and Gaussian noise.

system has to deal with more serious situations when classifying noisy images, such as the speed sign recognition for intelligent vehicles in bad weather (e.g., rain or snow)[59]. It is very interesting to verify the noise tolerance of the FTJ-based CNN system, and thus, the neural network was further trained by the F-MNIST database with different levels of salt & pepper noises and Gaussian noises. For salt & pepper noise, the

grayscales of some randomly chosen pixels were set to be white or black, and the ratio $\beta$ of chosen pixel number to the total pixel number is defined as noise level. While for Gaussian noise, noises that obey a Gaussian distribution with zero mean and standard deviation values $\zeta$ (ratio to the maximum pixel intensity, defined as noise level) were added to the images (see Supplementary S12)[60–62].

It can be seen from Fig. 7e that with increasing noise level, the recognition accuracy decreases. Interestingly, for the noisy F-MNIST dataset, the recognition accuracy still keeps >90% when salt & pepper noise level is 0.3 and Gaussian noise level is 0.2. And for the noisy MNIST dataset, the recognition accuracies can exceed 99% even when salt & pepper noise is 0.3 and Gaussian noise is 0.4, as shown in Supplementary Fig. S12. The robust fidelity in recognizing noisy images highlights the improvement in ferroelectric domain dynamics in the FTJ for practical applications in neuromorphic computing.

Our results provide an interesting strategy to reveal a prototype FTJ device with target performances for artificial synapses (Table S1)[14]. While for applications, the CMOS compatibility, such as the ease of fabrication of PZT on Si substrate, should be considered. Fortunately, according to the earlier reports[63,64], PZT films with a remnant polarization of 10 $\mu C/cm^2$ can be grown on Si at a low temperature of 400 °C. Besides, a transfer technique has been put forward to obtain PZT-based FTJ on Si substrate[65,66]. Especially, the $HfO_2$-based ferroelectric materials that can be grown on the Si substrate may be important to construct CMOS-compatible FTJs[67,68].

In summary, subnanosecond switching and numerous states were demonstrated in the FTJ with a (111)-oriented PZT barrier due to the ultrafine polydomain structure, the multiple ferro-electric switching dynamics and the low coercive voltages. As an artificial synapse, under a pulsed voltage of 10 ns, the FTJ shows high performance in terms of conductance manipulation with multiple states (256), low cycle-to-cycle variation (2.06%), sufficient dynamic range (~100×), long retention ($10^4$ s), good endurance ($10^9$) and linear (nonlinearity ~1) conductance manipulation, meeting the target specifications for synaptic devices. When the pulse duration decreases to 630 ps, as many as 150 states with a small cycle-to-cycle variation of 3.65% can still be achieved by applying voltage pulses ≤5 V. The fastest resistance switching speed in FTJs of 300 ps is achieved by voltages <13 V. Furthermore, the 50-nm-diameter FTJ shows an ultralow energy consumption of 5.3 fJ/bit. The CNN simulations based on the measured data of the FTJ device demonstrate a high recognition accuracy of 94.7% for F-MNIST images, close to that (~95.6%) of floating-point-based CNN software. The recognition accuracy higher than 90% for F-MNIST images can still be achieved even with 0.3 of salt & pepper noise or 0.2 of Gaussian noise. These results show the potential of (111)-oriented FTJs for constructing neuro-inspired computing systems.

## Methods

**Sample preparation**. Epitaxial PZT thin films (1.2 nm) were grown on (111)-oriented NSTO (0.7 wt% Nb) single-crystalline substrates by pulsed laser deposition (PLD) at a growth temperature of 525 °C under 200 mTorr in an $O_2$ atmosphere. The laser repetition rate and laser fluence were 1 Hz and 0.8 $J/cm^2$. After growth, the samples were cooled to 20 °C at a rate of 5 °C/min under 250 Torr in an $O_2$ atmosphere. The Pt, Ag and Cu top electrodes with a diameter of 100 μm were grown using magnetron sputtering with a shadow mask. For the nanoscale devices, 50-nm-diameter Pt top electrodes were patterned by electron-beam lithography and lift-off processes.

**Structural and ferroelectric characterizations**. A STEM system (JEM-ARM200F, JEOL, Japan) with a probe-forming spherical aberration corrector was utilized to investigate the cross-sectional structure of the PZT/NSTO. PFM studies were performed in scanning probe microscope (MFP-3D, Asylum Research, USA) using conductive tips (PPP-efm, Nanosensor, Switzerland).

**Real-time electrical measurements**. To ensure that the ultrafast voltage pulses can be delivered to the FTJ successfully, the high frequency circuit with a micro-strip waveguide was utilized (see detailed descriptions in Supplementary S13). And real-time measurements of subnanosecond pulse voltages were conducted to verify that the subnanosecond pulse voltages were successfully applied onto the FTJ devices (see in Supplementary S7). Subnanosecond pulse voltages were delivered by pulse generators (PSPL10300B, Tektronix, USA, or GZ1118GN-01EV/GZ1118GP-

01EV, Geozondas, Lithuania) to induce FTJ resistance switching. The FTJ resistance was read by an amperemeter (2410 SourceMeter, Keithley, USA) after operation voltage pulses were applied. The waveforms that passed through the device were monitored by an oscilloscope (DSA70804, Tektronix, USA). The DC and RF signals were separated by a switch matrix (RC-4SPDT-A18, 0–18 GHz, Mini-circuits, USA). Attenuators were connected before the oscilloscope to avoid overrange conditions. The scattering (S)-parameter were recorded by a vector network analyzer (AV3656A, CETC 41, China).

**Ferroelectric domain dynamics model**. The junction area is divided into two zones. One zone, $s_1$, undergoes a one-step switching process, while the other zone, $s_2$, undergoes a two-step switching process. The ferroelectric switching in each zone obeys a nucleation-limited-switching (NLS) model. The logarithm of the switching time of each zone, $\log(\tau_{sw})$, obeys a Lorentzian distribution[46].

$$F(\log \tau_{sw}) = \frac{s_i}{\pi} \left( \frac{w}{(\log \tau_{sw} - \log \tau_{mean})^2 + w^2} \right) \quad (2)$$

Here, $\tau_{mean}$ is the mean switching time, and $w$ and $\log(\tau_{mean})$ are the width and center of the distribution, respectively. The normalized summed switched area $s$ is as follows:

$$s = \sum_{i=1}^{2} s_i \left( \frac{1}{2} + \frac{1}{\pi} \arctan \frac{\log t_d - \log \tau_{mean}^i}{w_i} \right) \quad (3)$$

**Neural network simulations**. ResNet-18 CNN simulation based on the experimental device behavioral models were carried out to recognize F-MNIST and MNIST images. Noisy patterns were generated by adding Gaussian noise or salt & pepper noise to the pixels of the images. More details can be found in Supplementary S12.

**Reporting summary**. Further information on research design is available in the Nature Research Reporting Summary linked to this article.

## Data availability

All data supporting the findings of this study are available within the article and the Supplementary Information file. All data are available on request from the corresponding authors.

## Code availability

The software code used for this study are available on request from the corresponding authors.

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

## Acknowledgements

This work was supported by the National Key Research and Development Program of China (2019YFA0307900 and 2016YFA0300103), National Natural Science Foundation of China (51790491, U21A2066, 92163210, 51972296, and 52125204), and the fundamental research funds for the central universities (WK2030000035), and this work was partially carried out at the USTC Center for Micro and Nanoscale Research and Fabrication.

## Author contributions

Z.L. and Z.J.W. prepared the samples and performed the measurements. Z.L., H.W. and C.C.L. performed the ferroelectric domain dynamics analysis. Z.L. and H.Y.S. carried out the analysis of temperature dependent transport behaviors. Y.L. carried out the HAADF-STEM measurements. Z.Y.G., L.T.Z. and X.J. carried out the ResNet-18 simulations. X.G.L. and Y.W.Y. were responsible for the design of the overall experiments and

calculations. X.G.L. supervised the study. All the authors discussed the data and contributed to the manuscript.

## Competing interests

The authors declare no competing interests.
