## [Peer review file · Nature Communications]

REVIEWER COMMENTS

Reviewer #1 (Remarks to the Author):

The manuscript by Luo et al. demonstrated outstanding synaptic performances in ferroelectric tunnel junctions (FTJs) with subnanosecond, high-precision and linear weight updates. By introducing (111)-oriented PZT ultrathin films near the morphotropic phase boundary as ferroelectric barriers, the authors have achieved more synaptic states of 7~8 bits with ultrafast and affordable voltages. Especially, the switching speed can be as quick as ~300 ps for the writing and erasing of data. The performance improvement in the Ag/PZT/NbSTO tunnel junctions is, in my opinion, of high interest for both the fundamental and applied research communities. In addition, I find the simulation recognizing noisy images interesting because it goes closer to the practical situation and also distinguished from the previous results about the recognition simulations based on FTJs or resistive switching memories. The present work delivered important information and significant opportunities for the innovative development of ferroelectric tunneling devices. However, I also find several problems that should be addressed appropriately before this manuscript can be further considered to be accepted by Nature Communications.

1) The ferroelectricity in PZT is demonstrated by the STEM and PFM characterizations. However, the PFM phase mapping in Fig. 1d shows a smaller phase contrast <180 degree than the hysteresis loops in Fig. 1e. The authors should comment on this issue and give more convincing PFM results to show the robust ferroelectricity, for example the amplitude mapping and associated SKPFM results.

2) The authors mentioned that decreasing the ferroelectric film thickness would reduce the ferroelectric coercive voltage, and thus an ultrathin PZT ~1.2 nm was used as the barrier in the manuscript. However, it is known that the coercive field of the metal/ferroelectric/semiconductor FTJs may increase with decreasing thickness of the ferroelectric barrier because of the increase in the depleted region width of the semiconducting electrode, which also shares the applied voltage. Therefore, the relationship between the coercive voltage and thickness would be more complicated. The authors should give more systematical investigate on this issue.

3) The authors have conducted real-time current measurements during the application of the subnanosecond pulses. However, only the results under positive voltage were presented. Considering the non-symmetric structure and the rectification I-V of their FTJ, the results under negative voltages should be presented as well to confirm the subnanosecond speed for both voltage polarities. Furthermore, it will be more comprehensive to estimate the energy consumption by measuring the actual current during the negative voltage pulse as well.

4) The authors demonstrated a high endurance of 10^9 cycles in Fig. 4b. It is noted that the ON/OFF ratio starts declining after 10^8 cycles and is still sizable after 10^9 cycles. Did the FTJ damage or fatigued after 10^9 cycles? Please comment on it.

5) In Fig. 5, the FTJ shows very linear and stable conductance manipulations by varying operation pulses. As it is a highlighted result in the present work, could the authors compare this with reported memristors and give more insight on this issue?

Reviewer #2 (Remarks to the Author):

This manuscript by Luo et al. reports a ferroelectric tunnel junction based artificial synapse allowing high weight precision, good linearity and sub-nanosecond operation speed, which are desirable for neural network applications with high energy efficiency. The device parameters are extracted from electrical measurements and subsequently used for neural network simulations in software. Although the reported performance metrics at the device level are impressive, this reviewer has the following concerns regarding the manuscript.

1. The PZT based ferroelectric has long-standing issue in CMOS compatibility. The top and bottom electrode materials are also rarely used in CMOS processes. How to deal with such compatibility issue? If the electrode materials are replaced, will the device performance be affected?

2. The authors reported an ultrafast operation speed of 300-630 ps in the FTJ. Application of such short pulses usually requires sophisticated circuit design and special probes. Although Fig. S4 gives the waveforms of applied voltage and current response, it will be more convincing to verify that the parasitic effects in the circuit, i.e. RC, will actually allow such fast dynamics.
3. The present Ag/PZT/NSTO device shows reduced operation voltage after optimization. However, the voltage required is still 5 V (for 630 ps duration) and 13 V (for 300 ps duration). Such amplitudes are still too high for typical V_{dd} affordable in CMOS.
4. All the experimental data in the present study are from single devices without array demonstration, and the neural network simulations are based on simple MLP model and MNIST dataset. Such demonstration has been shown numerous times, which is no longer challenging and nothing significant. I suggest the authors at least show simulations on more complex models and more challenging datasets, to justify the advantage of the devices.

Reviewer #3 (Remarks to the Author):

The authors reported based on a ferroelectric tunnel junction based on a Ag/PbZr_{0.52}Ti_{0.48}O₃ (PZT, (111)-oriented)/Nb:SrTiO₃ stack structure and their switching performance. Based on such switching performance parameters, they further performed simulations to predict the performance of such devices in a neural network to classify MNIST dataset. The results are interesting and useful. The following questions need to be addressed before considering publishing in Nature Communications.

1. The authors refer to previous works stating that reducing the ferroelectric barrier thickness generally results in polydomains. Was the same phenomenon observed and verified with PZT for this work? And was the thickness of the ferroelectric thin film optimized this way?
2. What was the motivation behind using a highly diffusive metal like silver as one of the electrodes? How to prevent the migration of Ag to form a diffusive memristor like what reported in the literature (see Nature materials 16 (1), 101) Considering the ultrathin nature of the ferroelectric film would not a metal like Platinum with work function greater than silver serve the purpose better?
3. What is the grain size of the ferroelectric film at 1.2nm thickness? With 100um wide silver electrode, one would expect to capture a few grain boundaries which could prove detrimental for the device endurance.
4. In the ferroelectric domain switching dynamics section, the authors write that the FTJ was first SET to ON state but at same time use V_{RESET} to label the voltage pulse. This appears confusing and contradicting. Usually, it is referred to as V_{SET} in memristive devices nomenclature. Since, the authors also refer to some of the works on memristive devices it would make more sense if they made this change.
5. Endurance of this device is exceptional with ON/OFF ratio degrading very little over 10^9 cycles. It would be interesting to study its dependence on top electrode, for example Silver versus Platinum which can make the FJT more robust. It would be appreciated by the research community if results of such experiment were shared.
6. For sub-nanosecond pulse measurement, usually a coplanar waveguide device structure is made to ensure the ultra-fast pulse maintains its shape when it is delivered to the device (see Advanced Functional Materials 26 (29), 5290-5296). How did the authors ensure their sub-ns measurements?
7. High Temperature and PLD were used to obtained high quality of epitaxial growth of the ferroelectric film. Can the authors comment on the possibility of using more industry-friendly-tools, such as sputtering, to prepare such films? How would the FTJ performance of sputtering deposited films compare with the PLD deposited devices?
8. Please clearly list what properties were measured on the 100um devices and 80nm devices, respectively.

A list of changes

Main Text:

1. In **Line 3 of Page 1** of the revised manuscript, we all agree to add “Zeyu Guan” to be the third author because of his contributions on the ResNet-18 simulations.
2. In **Lines 16-18 of Page 1**, the sentence “Based on the experimental performance.....floating-point-based CNN software.” was added.
3. In **Lines 40-41 of Page 2**, the phrase “and more complicated fashion product images in the Fashion-MNIST (F-MNIST) database” was added.
4. In **Lines 94-95 of Page 5**, the phrase “~9.24 fJ for an 80-nm FTJ” was replaced by “~5.3 fJ for a 50-nm-diameter FTJ.”
5. In **Lines 96-101 of Page 5**, the sentences “Based on the performance of the experimental device..... neuromorphic computing.” were added.
6. In **Line 124 of Page 6 to Line 136 of Page 7**, the sentences “and scanning Kelvin Probe Microscopy (SKPM)..... of the ultrathin PZT film.” were added.
7. In **Lines 142-153 of Page 8**, the sentences “The evolution..... in FTJ devices.” were added.
8. In **Line 176 of Page 9**, the “ V_{reset} ” was replaced by V_{set} .
9. In **Line 228 of Page 12 to Line 246 of Page 13**, the sentences “For our FTJs..... (see detailed descriptions in the Supplementary S6).” were added.
10. In **Lines 249-251 of Page 13**, the sentence “As a result..... as shown in Supplementary S7.” was added.
11. In **Lines 252-253 of Page 13 and Lines 360-361 of Page 19**, the “80 nm” was replaced by “50 nm”, and the “9.24 fJ/bit” was replaced by “5.3 fJ/bit”.
12. In **Lines 268-273 of Page 14**, the sentences “In addition..... (see Supplementary Fig. S10).” were added.
13. In **Lines 284-289 of Page 15**, the sentence “Here, the variable voltage..... standard deviation of each conductance state).” was added.
14. In **Lines 313-332 of Page 17**, the sentences “As shown in Fig. 7a.....with different levels of salt & pepper noises and Gaussian noises.” were added.
15. In **Lines 337-343 of Page 18**, the sentences “It can be seen from..... in neuromorphic computing.” were added.
16. In **Lines 344-350 of Page 18**, the sentences “Our results provide..... important to construct CMOS-compatible FTJs.” were added.

17. In **Lines 361-364 of Page 19**, the sentences “The CNN simulations..... with 0.3 of salt & pepper noise or 0.2 of Gaussian noise.” were added.
18. In **Lines 379-381 of Page 20**, the sentence “To ensure that the ultrafast voltage pulses..... (see detailed descriptions in Supplementary S13)” was added.
19. In **Line 390-391 of Page 20**, the sentence “The scattering (S)-parameter were recorded by a vector network analyzer (AV3656A, CETC 41, China).” was added.
20. In **Line 401-404 of Page 21**, the sentences “ResNet-18 CNN simulation..... More details can be found in Supplementary S12.” were added.
21. In the revised manuscript, **Code availability** was added in **Lines 408-410 of page 21**.
22. In the revised manuscript, **Figs. 1, 3, 4, 5, and 7** were replotted. **Fig. 2** was added.

Supplementary information:

1. In the Supplementary information, **S4, S5, S6, S10, S11 and S13** were added.
2. In **S7, S8, S9 and S12** of Supplementary information, the descriptions were rewritten.

There are also some minor revisions about the sentences to improve English language as well as several reference updates, which are not listed here.

Responses to Reviewer #1

Thank you very much for your valuable suggestions and pertinent comments on our manuscript. We have made careful revisions following your suggestions. We wish that the revised version and the responses would be satisfactory to you.

Comment 1. *The ferroelectricity in PZT is demonstrated by the STEM and PFM characterizations. However, the PFM phase mapping in Fig. 1d shows a smaller phase contrast <180 degree than the hysteresis loops in Fig. 1e. The authors should comment on this issue and give more convincing PFM results to show the robust ferroelectricity, for example the amplitude mapping and associated SKPFM results.*

Answer 1: We thank the reviewer for pointing out this important issue. The piezoelectric force microscopy (PFM) phase contrast in Fig. 1d of previous version is indeed slightly smaller than 180 degree, and this should be due to the non-optimized parameter settings on frequency difference, *etc.* during dual AC resonance tracking (DART) mode PFM measurements. Following your advice, further PFM and scanning Kelvin probe microscopy SKPM measurements have been carried out to confirm the robust ferroelectricity in our 1.2 nm-thick (111)-oriented $\text{PbZr}_{0.52}\text{Ti}_{0.48}\text{O}_3$ (PZT) film, as follows:

1) PFM phase and amplitude mappings. Further PFM mappings were carefully carried out after the box-in-box ferroelectric domain patterning by reversed tip biases (-6 V and +5 V). As the PFM phase mapping shown in Fig. RI-1a, the phase contrast between poled-up and poled-down regions reaches 180 degree. And the domain boundaries can be clearly seen in the PFM amplitude diagram. These evidence the reversible ferroelectric domains in PZT ultrathin film.

2) SKPM mapping. The SKPM surface potential mapping associated with the above PFM mappings was also tested, as shown in Fig. RI-1c. It can be seen that the surface potential in the poled-up region is lower than that of the poled-down region. This is due to that the polarization charges are overcompensated by screening charges injected from tips, consistent with previous SKPM results on PZT and BiFeO_3 ferroelectric films¹⁻³.

3) PFM hysteresis loops. The deformed PFM hysteresis loop measured with a large AC voltage is a strong evidence of ferroelectricity⁴. As shown in Fig. RI-1d and e, the PFM phase and amplitude hysteresis loops were characterized with different AC

voltages. The loops deform with increasing AC voltages, indicating the robust ferroelectricity of the ultrathin PZT films in our FTJs⁵.

Fig. RI-1 | Ferroelectric characterizations by PFM and SKPM. a PFM phase, **b** PFM amplitude, and **c** SKPM surface potential images recorded after writing an area of $3 \times 3 \mu\text{m}^2$ with -6 V and the central $1.5 \times 1.5 \mu\text{m}^2$ with $+5 \text{ V}$ on the (111)-oriented PZT (1.2 nm)/NSTO. **d** PFM phase and **e** amplitude loops collected with various AC voltages.

Fig. RI-1a ~ e is added as **Fig. 1c ~ g**. The corresponding discussions were added in **Line 124 of Page 6** to **Line 136 of Page 7** of the revised manuscript.

Comment 2. *The authors mentioned that decreasing the ferroelectric film thickness would reduce the ferroelectric coercive voltage, and thus an ultrathin PZT $\sim 1.2 \text{ nm}$ was used as the barrier in the manuscript. However, it is known that the coercive field of the metal/ferroelectric/semiconductor FTJs may increase with decreasing thickness of the ferroelectric barrier because of the increase in the depleted region width of the semiconducting electrode, which also shares the applied voltage. Therefore, the relationship between the coercive voltage and thickness would be more complicated. The authors should give more systematical investigate on this issue.*

Answer 2: Thank you for the great suggestion. To figure out the relationship between coercive voltage (V_c) of FTJ and thickness (d) of ferroelectric film, the resistance vs. pulsed voltage loops of FTJs with different ferroelectric film thicknesses were measured at pulse durations of $t_d = 10 \text{ ns}$ and 630 ps , as shown in Fig. RI-2a and b, respectively. The coercive voltages V_c can be calculated through averaging the absolute values of positive and negative coercive voltages in Fig. RI-2a and b. The values of V_c

decrease with decreasing PZT thickness d from 6.0 nm to 1.2 nm, and the relationship between V_c and d can be fitted by a power law:

$$V_c \propto d^\alpha. \quad (\text{RI-1})$$

As shown in Fig. RI-2c and d, the fitted α are 0.30 for $t_d = 10$ ns and 0.26 for $t_d = 630$ ps, respectively. This is very close to the Kay-Dunn law $V_c = d^{1/3}$ for ferroelectric thin films⁶, such as $\alpha = 0.38$ for PZT⁷ and $\alpha = 0.331$ for BiFeO₃⁸. The slightly smaller α than that $\sim 1/3$ of Kay-Dunn law⁹ may be related to the depleted region at the semiconductor interface which also shares the applied voltage as the reviewer pointed out.

Fig. RI-2 | Ferroelectric film thickness dependence of coercive voltage and resistance for FTJs. a, b Resistance vs. pulsed voltage with pulse durations of $t_d = 10$ ns and $t_d = 630$ ps, respectively. **c, d** Thickness dependent coercive voltage with $t_d = 10$ ns and $t_d = 630$ ps, respectively. The solid lines are the fitting results by a power law $V_c \propto d^\alpha$. **e, f** Thickness dependent resistances for ON and OFF states with $t_d = 10$ ns and $t_d = 630$ ps, respectively. The solid lines are the fitting results by an exponential law $R \propto \exp(d)$.

Interestingly, it is noted that with increasing PZT film thickness, the FTJ resistance increases. As shown in Fig. RI-2e and f, the extracted resistances R for both ON and OFF states from Fig. RI-2a and b clearly show exponential dependences on d , i.e., $R \propto \exp(d)$, which indicates the tunneling effect in FTJs¹⁰.

Fig. RI-2 is added as **Supplementary Fig. S3**. The relevant descriptions have been added in **Lines 228-235 of Page 12** of the revised manuscript and **S4** of the revised Supplementary information.

Comment 3. *The authors have conducted real-time current measurements during the application of the subnanosecond pulses. However, only the results under positive voltage were presented. Considering the non-symmetric structure and the rectification I-V of their FTJ, the results under negative voltages should be presented as well to confirm the subnanosecond speed for both voltage polarities. Furthermore, it will be more comprehensive to estimate the energy consumption by measuring the actual current during the negative voltage pulse as well.*

Answer 3: Thank you for pointing out this important issue. Following your suggestion, the real-time current measurement under negative voltage pulse was carried out, as shown in Fig. RI-3. By applying a negative voltage pulse of ~600 ps shown in Fig. RI-3a, the transmitted current signal captured by the oscilloscope is depicted in Fig. RI-3b. It can be seen that the transmitted signal is deformed in shape with an even smaller pulse width ~470 ps (defined as full width at half maximum). The write current density is about 1.3×10^3 A/cm², and the energy consumption can be estimated to be ~520 pJ per operating pulse. Here, the different deformed current signals between positive and negative voltage pulses are related to the rectification characteristic of our FTJ as pointed out by the reviewer, and this will lead to the different impedance mismatching conditions. Similar deformed transmitted signals have also been reported in other memristors^{11,12}.

Fig. RI-3 is added in **Supplementary Fig. S6**. The relevant description is added in **Lines 249-251 of Page 13** of the revised manuscript and **S7** of the revised Supplementary information.

Fig. RI-3 | Real-time electrical measurement under a negative voltage pulse. a A voltage pulse of 600 ps and -11 V applied to the FTJ top electrode. **b** The signal transmitted through the FTJ was recorded by an oscilloscope.

Comment 4. *The authors demonstrated a high endurance of 10^9 cycles in Fig. 4b. It is noted that the ON/OFF ratio starts declining after 10^8 cycles and is still sizable after 10^9 cycles. Did the FTJ damage or fatigued after 10^9 cycles? Please comment on it.*

Answer 4: As shown in Fig. RI-4 in the manuscript, there is a gradual fatigue process from $\sim 10^8$ cycles to $\sim 10^9$ cycles. In fact, after applying 10^9 cycles of +2 V/-3 V voltage pulses by a pulse generator, another 5 cycles of resistance switchings were measured as shown in Fig. RI-4b. It can be seen that the FTJ was not damaged but fatigued with the ON/OFF ratio decreasing to ~ 140 .

Fig. RI-4 | Resistance switching measurement after 10^9 cycles. a Endurance of 10^9 cycles by applying +2 V/-3 V voltage pulses with $t_d = 100$ ns, **b** Resistance switching measurements after the 10^9 cycles.

Fig. RI-4b is added as **Fig. 5c**. The relevant description is added in **Lines 268-271** of **Page 14** of the revised manuscript.

Comment 5. *In Fig. 5, the FTJ shows very linear and stable conductance manipulations by varying operation pulses. As it is a highlighted result in the present*

work, could the authors compare this with reported memristors and give more insight on this issue?

Answer 5: We deeply appreciate the reviewer's suggestion. Following your suggestion, the representative performances of different types of memristors including conducting filament modulated memristors, interfacial ionic displacement type memristors, ferroelectric field effect transistor (FeFET) memristors, and FTJ memristors were listed in Table RI-1.

Table RI-1 Overview of representative performances of different memristors¹³⁻¹⁹

Device Type \ Metrics	Mechanism	Nonlinearity (Set/Reset)	Dynamic range	Cycle variation	State number	Operation voltage
Ta/HfO _x /Pt ¹³	Filament modulation	0.1 / -0.1	10	2%	100	0.6 ~1.6 V @ 10 μs
TEL/HfO _x ¹⁴	Filament modulation	0.96/-3.26	10	<3%	128	1.6 V/-1.6 V @ 50 ns
TiN/PCMO/Pt ¹⁵	Interfacial	-3.4/-2.5	100	<2%	100	-1.5~ -3.5 V /0.5~2.5 V @ 100 μs
Pt/AlO _x N-rich TiN/PCMO/Pt ¹⁶	Interfacial	3.68/-6.76	6.8	<1%	50	3 V/-3 V @ 1 ms
TiN/HZO/SiO ₂ /Si FeFET ¹⁷	FeFET	1.75/1.46	45	<0.5%	32	2.85~4.45 V/-2.1~ -3.8 V @ 75 ns
TiN/HZO/SiO ₂ /Si FeFET ¹⁷	FeFET	5.54/-8.08	8	–	20	3.7 V/-3.2 V @ 75 ns
Pt/BTO/Nb:STO ¹⁸	FTJ	–	10	2.5%	200	1.3 V/-1.75 V @ 50 ns
Au/PVDF/Nb:STO ¹⁹	FTJ	1.2/1.6	2.2	–	75	2 V/-7 V @ 20 ns
Our work	FTJ	0.77/-0.94	100	2.06%	256	1.35~2 V/-1.4~ -3.5 V @ 10 ns
		1.20/-1.09	30	3.65%	150	3.5~4.5 V/-3.2~ -5 V @ 630 ps

It is noted that the conductance manipulations operated by variable voltages are usually more linear than using invariable voltages. Thus, variable voltage scheme is chosen for conductance manipulation of our devices. Based on the continuous ferroelectric switching dynamic of (111)-oriented PZT (~1.2 nm) of FTJs, conductance manipulation of our device is very linear (nonlinearity ~1), satisfying the targeted specification for a synaptic device²⁰. Besides the outstanding linearity, our device

shows comprehensive performance advantages with high-precision (256 states, 8 bits), reproducible (cycle-to-cycle variation ~2.06%), symmetric weight updates and subnanosecond operation speed.

Table RI-1 is added as **Table S3**. The relevant description is added in **lines 284-286 of Page 15** of the revised manuscript and **S11** of the revised Supplementary information.

References

- 1 Fan, Z., Fan, H., Lu, Z. X., Li, P. L., Huang, Z. F., Tian, G., Yang, L., Yao, J. X., Chen, C., Chen, D. Y., Lu, X. B., Gao, X. S. & Liu, J. M. Ferroelectric diodes with charge injection and trapping. *Phys. Rev. Appl.* **7**, 014020 (2017).
- 2 Tan, Z., W., Hong, L. Q., Fan, Z., Tan, J. J., Zhang, L. Y., Jiang, Y., Hou, Z. P., Chen, D., Y., Qin, M. H., Zeng, M., Gao, J. W., Lu, X. B., Zhou, G. F., Gao, X. S. & Liu, J. M. Thinning ferroelectric films for high efficiency photovoltaics based on the Schottky barrier effect. *NPG Asia Mater.* **11**, 20 (2019).
- 3 Ren, Z. Q., Liu, Y. D., Bao, S. Y., Yang, N., Zhong, N., Tang, X. D., Xiang, P. H. & Duan, C. G. Probing the origins of electroresistance switching behavior in ferroelectric thin films. *Appl. Phys. Lett.* **115**, 242901 (2019).
- 4 Balke, N., Maksymovych, P., Jesse, S., Herklotz, A., Tselev, A., Eom, C. B., Kravchenko, I. I., Yu, P. & Kalinin, S. V. Differentiating ferroelectric and nonferroelectric electromechanical effects with scanning probe microscopy. *ACS Nano* **9**, 6484-6492 (2015).
- 5 Xi, Z. N., Ruan, J. J., Li, C., Zheng, C. Y., Wen, Z., Dai, J. Y., Li, A. D. & Wu, D. Giant tunnelling electroresistance in metal/ferroelectric/semiconductor tunnel junctions by engineering the Schottky barrier. *Nat. Commun.* **8**, 15217 (2017).
- 6 Xu, R. J., Gao, R., Reyes-Lillo, S. E., Saremi, S., Dong, Y. Q., Lu, H. L., Chen, Z. H., Lu, X. Y., Qi, Y. J., Hsu, S. L., Damodaran, A. R., Zhou, H., Neaton, J. B. & Martin, L. W. Reducing coercive-field scaling in ferroelectric thin films via orientation control. *ACS Nano* **12**, 4736-4743 (2018).
- 7 Bjormander, C., Sreenivas, K., Duan, M., Grishin, A. M. & Rao, K. V. Thickness dependence of the coercive electric field of laser ablated niobium-doped lead-zirconium-titanate films. *Appl. Phys. Lett.* **66**, 2493, (1995).
- 8 Steffes, J. J., Ristau, R. A., Ramesh, R. & Huey, B. D. Thickness scaling of ferroelectricity in BiFeO₃ by tomographic atomic force microscopy. *Proc. Natl.*

- Acad. Sci. U.S.A.* **116**, 2413-2418 (2019).
- 9 Kay, H. F. & Dunn, J. W. Thickness dependence of the nucleation field of triglycine sulphate. *Philos. Mag.* **7**, 2027-2034 (1962).
 - 10 Li, C. J., Huang, L. S., Li, T., Lv, W. M., Qiu, X. P., Huang, Z., Liu, Z. Q., Zeng, S. W., Guo, R., Zhao, Y. L., Zeng, K. Y., Coey, M., Chen, J. S., Ariando & Venkatesan, T. Ultrathin BaTiO₃-Based ferroelectric tunnel junctions through interface engineering. *Nano Lett.* **15**, 2568-2573 (2015).
 - 11 Torrezan, A. C., Strachan, J. P., Medeiros-Ribeiro, G. & Williams, R. S. Sub-nanosecond switching of a tantalum oxide memristor. *Nanotechnology* **22**, 485203 (2011).
 - 12 Wang, C., Wu, H. Q., Gao, B., Wu, W., Dai, L. J., Li, X. Y. & Qian, H. Ultrafast RESET analysis of HfO_x-Based RRAM by sub-nanosecond pulses. *Adv. Electron. Mater.* **3**, 1700263 (2017).
 - 13 Li, C., Belkin, D., Li, Y. N., Yan, P., Hu, M., Ge, N., Jiang, H., Montgomery, E., Lin, P., Wang, Z. R., Song, W. H., Strachan, J. P., Barnell, M., Wu, Q., Williams, R. S., Yang, J. J. & Xia, Q. F. Efficient and self-adaptive in-situ learning in multilayer memristor neural networks. *Nat. Commun.* **9**, 2385 (2018).
 - 14 Wu, W., Wu, H. Q., Gao, B., Deng, N., Yu, S. M. & Qian, H. Improving analog switching in HfO_x-Based resistive memory with a thermal enhanced Layer. *IEEE Electron. Device Lett.* **38**, 1019-1022 (2017).
 - 15 Jang, J. W., Park, S. S., Burr, G. W., Hwang, H. S. & Jeong, Y. H. Optimization of conductance change in Pr_{1-x}Ca_xMnO₃-Based synaptic devices for neuromorphic systems. *IEEE Electron. Device Lett.* **36**, 457-459 (2015).
 - 16 Park, S., Sheri, A., Kim, J., Noh, J., Jang, J., Jeon, M., Lee, B., Lee, B. R., Lee, B. H. & Hwang, H. Neuromorphic speech systems using advanced ReRAM-based synapse. *International Electron Devices Meeting (IEDM)*, 25.6.1-25.6.4 (IEEE, Washington, DC, USA, 2013).
 - 17 Jerry, M., Chen, P. Y., Zhang, J. C., Sharma, P., Ni, K., Yu, S. M. & Datta, S. Ferroelectric FET analog synapse for acceleration of deep neural network training *International Electron Devices Meeting (IEDM)*, 6.2.1-6.2.4 ((IEEE, San Francisco, CA, USA, 2017)).
 - 18 Li, J. K., Ge, C., Du, J. Y., Wang, C., Yang, G. Z. & Jin, K. J. Reproducible ultrathin ferroelectric domain switching for high-performance neuromorphic computing. *Adv. Mater.* **32**, e1905764 (2020).

- 19 Majumdar, S. Ultrafast switching and linear conductance modulation in ferroelectric tunnel junctions via P(VDF-TrFE) morphology control. *Nanoscale* **13**, 11270-11278 (2021).
- 20 Xi, Y., Gao, B., Tang, J. S., Chen, A., Chang, M. F., Hu, X. B. S., Van Der Spiegel, J., Qian, H. & Wu, H. Q. In-memory learning with analog resistive switching memory: a review and perspective. *Proc. IEEE* **109**, 14-42 (2020).

Responses to Reviewer #2

Thank you very much for your pertinent comments and valuable suggestions on our manuscript. We have carefully revised the manuscript following your suggestions, and our responses to your comments are listed point by point as follows:

Comment 1. *The PZT based ferroelectric has long-standing issue in CMOS compatibility. The top and bottom electrode materials are also rarely used in CMOS processes. How to deal with such compatibility issue? If the electrode materials are replaced, will the device performance be affected?*

Answer 1: We sincerely thank the reviewer for raising this important topic. Our current work focuses on providing an interesting strategy to reveal a prototype FTJ device with target performances for artificial synapses. Here, PZT was chosen as the FTJ barrier because of its low coercive field and robust ferroelectricity. Indeed, as the reviewer pointed out, how to integrate $\text{PbZr}_{0.52}\text{Ti}_{0.48}\text{O}_3$ (PZT) on Si substrate is still challenging. Fortunately, there are amount of reports devoted to solving this issue^{1,2}. For example, Bretos et al. reported that the ferroelectric PZT films with a remnant polarization of $10 \mu\text{C}/\text{cm}^2$ can be grown on Si at a low temperature of 400°C , showing the potential of PZT for CMOS compatibility². On the other hand, the HfO_2 -based ferroelectric materials that can be grown on the Si substrate would be an important solution to construct CMOS-compatible FTJs^{3,4}. However, it is still challenging to obtain pure ferroelectric phase and reduce coercive field.^{5,6}

As for electrodes, FTJ devices with other electrodes including Cu (CMOS compatible⁷) and Pt were fabricated. As shown in Fig. RII-1a, b and c, the resistance vs. pulsed voltage loops with different electrodes were measured by using voltage pulses of $t_d = 100 \text{ ns}$, 10 ns , and 630 ps , respectively. It can be seen that by varying electrode from Ag, Cu to Pt, the work functions of electrode increase, resulting in a higher ON/OFF ratio, larger resistance, and bigger coercive voltage, which is consistent with the earlier report⁸. Especially, it is noted that the FTJ with CMOS-compatible Cu electrode can also be operated at subnanosecond but with a higher voltage. In principle, other CMOS-compatible electrodes with lower work function (such as TiN, W) would be preferred in terms of the operating voltage. Similar to PZT, the bottom electrode $\text{Nb}:\text{SrTiO}_3$ (NSTO) is also difficult for CMOS compatibility. However, it is worth mentioning that very recently, researchers have developed a transfer technique for obtaining perovskite films on Si substrate⁹, and

PZT-based FTJ that is transferred on Si with a large ON/OFF ratio ~ 1000 was demonstrated¹⁰. This may be feasible for realizing PZT-based FTJs with CMOS compatibility.

Fig. RII-1 | FTJs with different electrodes. Resistance vs. pulsed voltage loops with different top electrodes with **a** $t_d = 100$ ns, **b** $t_d = 10$ ns, and **c** $t_d = 630$ ps, respectively.

Fig. RII-1 is added as **Supplementary Fig. S4**. The relevant description is added in **Line 235 of Page 12 to Line 240 of Page 13** and **Lines 344-350 of Page 18** of the revised manuscript and **S5** of the revised Supplementary information.

Comment 2. *The authors reported an ultrafast operation speed of 300-630 ps in the FTJ. Application of such short pulses usually requires sophisticated circuit design and special probes. Although Fig. S4 gives the waveforms of applied voltage and current response, it will be more convincing to verify that the parasitic effects in the circuit, i.e. RC, will actually allow such fast dynamics.*

Answer 2: We are very grateful for the reviewer's suggestion. To carry out the ultrafast measurements, the FTJ was connected to the high frequency circuit through a microstrip waveguide¹¹ by short copper wires. The characteristic impedance $Z_0 = 50 \Omega$ of the microstrip waveguide matches with the connection cables (0-18GHz, Mini-circuits, USA), pulse generator (PSPL10300B, Tektronix, USA), oscilloscope (DSA70804, Tektronix, USA) and switch matrix (RC-4SPDT-A18, 0-18 GHz, Mini-circuits, USA). And the connection wire between the FTJ and waveguide is as short as possible. To verify the parasitic effect in the circuit, the high frequency scattering (S)-parameter measurements up to 3 GHz were performed by using a vector network analyzer (VNA) (AV3656A, CETC 41, China). The measured S-parameter with ON and OFF states are shown in Fig. RII-2a-d. Here, for the phase data in Fig. RII-2b, d, the phase shift contributed from the microstrip lines has already been subtracted¹².

Fig. RII-2 | Scattering parameter characterized by a vector network analyzer. a Magnitude and **b** phase of transmission and reflection of the FTJ measured at OFF state, **c** magnitude and **d** phase of reflection and transmission of the FTJ measured at ON state. **e** The equivalent circuit model used for the simulations.

An equivalent circuit model is used to analyze the S-parameter results, similar to the previous work¹², as shown in Fig. RII-2e. Here, the C_m is the parasitic capacitance of memristor, R_s is the series contact resistance between the FTJ and waveguide, and R_m is the parallel resistance that is approximately equal to the FTJ resistance. Similar to the earlier reports^{13,12}, the S-parameters of transmission (V_{trans}/V_{inc}) and reflection (V_{refl}/V_{inc}) can be calculated by Equation RII-1 and Equation RII-2, respectively:

$$\frac{V_{trans}}{V_{inc}} = \left(\frac{2Z_0}{2Z_0 + Z_L} \right), \quad (\text{RII-1})$$

$$\frac{V_{refl}}{V_{inc}} = \left(\frac{Z_L}{2Z_0 + Z_L} \right), \quad (\text{RII-2})$$

where $Z_0 = 50 \Omega$ is the characteristic impedance and Z_L is the impedance of the equivalent circuit that was calculated by advanced design system (ADS) software. The results in Fig. RII-2a-d can be described nicely by the equivalent circuit model simulated using the advanced design system (ADS) software. For OFF state, the R_s is $\sim 36 \Omega$, C_m is ~ 4.8 pF, and R_m is approximately equal to $10^6 \Omega$. Thus, the RC delay can be estimated to be $R_s \times C_m = 172$ ps¹⁴. While for ON state, the R_s is $\sim 30 \Omega$, C_m is ~ 5.1 pF, R_m is approximately equal to $10^4 \Omega$, and the RC delay can be estimated to be 153 ps. It can be seen that the RC delays are shorter than the widths of applied pulses, showing that subnanosecond pulse signal can be applied to the FTJ devices successfully.

Fig. RII-2 is added as **Supplementary Fig. S14**. The relevant description is added in **Methods** of the revised manuscript and **S13** of the revised Supplementary information.

Comment 3. *The present Ag/PZT/NSTO device shows reduced operation voltage after optimization. However, the voltage required is still 5 V (for 630 ps duration) and 13 V (for 300 ps duration). Such amplitudes are still too high for typical V_{dd} affordable in CMOS?*

Answer 3: Thank you for raising this important topic. As you pointed out, our FTJ shows reduced operation voltage after optimization, with the fastest switching speed among the reported FTJs^{4,8}. Although the voltages of 5-13 V are larger than typical V_{dd} ¹⁵, it is lower than the operation voltage of NAND flash which is usually around 20 V¹⁶. Therefore, similar to the NAND flash, a charge pump circuit may be used to realize affordable operation voltages for ultrafast FTJs¹⁷.

The relevant description is added in **S9** of the revised Supplementary information.

Comment 4. *All the experimental data in the present study are from single devices without array demonstration, and the neural network simulations are based on simple MLP model and MNIST dataset. Such demonstration has been shown numerous times, which is no longer challenging and nothing significant. I suggest the authors at least show simulations on more complex models and more challenging datasets, to justify the advantage of the devices?*

Answer 4: Thanks for your great suggestion. Following your suggestion, more complicated convolutional neural network (CNN) ResNet-18¹⁸ was used to recognize

the more challenging fashion product images in Fashion (F)-MNIST dataset¹⁹ with different levels of salt & pepper and Gaussian noises, as shown in Fig. RII-4. The ResNet-18 neural network contains 17 convolution layers and a final full connection layer. For the first convolution layer, the size of convolution kernel is 5×5 . Afterwards, there are 8 blocks, and each block contains 2 convolutional layers. The sizes of convolution kernels for these blocks are all 3×3 . The number of kernels increases from 64 for B1, 128 for B2, 256 for B3, to 512 for B4 blocks. And the feature size decreases from 32 for B1, 16 for B2, 8 for B3, to 4 for B4 block. The input image data converted from each pixel grayscale will flow through each convolution block by two ways simultaneously. In one way, the data are convolved and flow through the convolution block. In the other way, the data are transmitted around the convolution block by a shortcut connection. These two types of data are summed after each block and transmitted through blocks one by one. In addition, there are two kinds of shortcut connections. In the shortcut connections for B1_1, B1_2, B2_2, B3_2 and B4_2 convolutional layers, data are transmitted directly and are plotted as solid arrows in Fig. RII-4a. While in the shortcut connections for B2_1, B3_1 and B4_1 convolutional layers, data are convolved by 1×1 convolution kernels and are plotted as dotted arrows in Fig. RII-4a. The last convolutional block B4_2 is followed by the final full connection layer with 10 output neurons.

To justify the advantage of the FTJ devices, the CNN simulations were carried out based on the experimentally obtained behavior model including 256 states with 10 ns pulse duration (see Fig. S11a) or 150 states with 630 ps pulse duration (see Fig. S11b). The simulation results are shown in Fig. RII-4b. It can be seen that the CNN simulation based on 256 states shows a higher recognition accuracy of 94.7% for F-MNIST, which is close to that achieved by floating-point-based software (95.6%), demonstrating the excellent performance of our FTJ synapses. When the simulation was performed based on 150 states, the recognition accuracy decreases, but is still as high as 90.0%. Besides, F-MNIST database with different levels of salt & pepper noises and Gaussian noises were also used to train the CNN network to study noise tolerance. It can be seen that with increasing noise level, the recognition accuracy decreases. Interestingly, the recognition accuracy can maintain $>90\%$ when salt & pepper noise is 0.3 and Gaussian noise is 0.2. These more complicated simulation works further evidence the advantages of FTJs for neural network based neuromorphic computing.

Fig. RII-4 | CNN simulation for F-MNIST images. **a** Schematic diagram of the ResNet-18 neural network. **b** Simulation results based on the experimental results with 256 (in Fig. 5a) and 150 (in Fig. 5b) conductance states and floating-point-based software. **c, d** Training results with different levels of salt & pepper noise and Gaussian noise. **e** Recognition accuracy of F-MNIST images with different levels of salt & pepper noise and Gaussian noise.

In addition, the CNN simulations on common MNIST dataset were also carried out. As shown in Fig. RII-5, high accuracies of 99.7%, 99.5% and 99.1% were achieved in the CNN simulations based on floating point, 256 states and 150 states,

respectively. The recognition can be >99% even when salt & pepper noise is 0.3 and Gaussian noise is 0.4, demonstrating the high performance of FTJs.

Fig. RII-5 | CNN simulation for MNIST digits. **a** Simulation results based on the experimental results with 256 (in Fig. 6a) and 150 (in Fig. 6b) conductance states and floating-point-based software for MNIST images. **b, c** Training results with different levels of Gaussian noise and salt & pepper noise. **d** Recognition accuracy of MNIST images with different levels of Gaussian noise and salt & pepper noise.

Fig. RII-4 is added as **Fig. 7**. **Fig. RII-5** is added as **Fig. S12**. The relevant description is added in **Lines 313-332 of Page 17, Lines 337-343 of Page 18** of the revised manuscript, and **S12** of the revised Supplementary information. Simulation results of simple MLP model on MNIST dataset in previous manuscript were replaced by the complicated ResNet-18 network simulation on F-MNIST dataset.

References

- 1 Ghoneim, M. T., Zidan, M. A., Alnassar, M. Y., Hanna, A. N., Kosel, J., Salama, K. N. & Hussain, M. H. Thin PZT-Based ferroelectric capacitors on flexible silicon for nonvolatile memory applications. *Adv. Electron. Mater.* **1**, 1500045 (2015).
- 2 Bretos, I., Jiménez, R., Tomczyk, M., Rodríguez-Castellón, E., Vilarinho, P. M. & Calzada, M. L. Active layers of high-performance lead zirconate titanate at temperatures compatible with silicon nano- and microelectronic devices. *Sci. Rep.* **6**, 20143 (2016).

- 3 Cheema, S. S., Kwon, D., Shanker, N., dos Reis, R., Hsu, S. L., Xiao, J., Zhang, H. G., Wagner, R., Datar, A., McCarter, M. R., Serrao, C. R., Yadav, A. K., Karbasian, G., Hsu, C. H., Tan, A. J., Wang, L.C., Thakare, V., Zhang, X., Mehta, A., Karapetrova, E., V Chopdekar, R., Shafer, P., Arenholz, E., Hu, C. M., Proksch, R., Ramesh, R., Ciston, J. & Salahuddin, S. Enhanced ferroelectricity in ultrathin films grown directly on silicon. *Nature* **580**, 478-482 (2020).
- 4 Berdan, R., Marukame, T., Ota, K., Yamaguchi, M., Saitoh, M., Fujii, S., Deguchi, J. & Nishi, Y. Low-power linear computation using nonlinear ferroelectric tunnel junction memristors. *Nat. Electron.* **3**, 259-266 (2020).
- 5 Yang, H., Lee, H. J., Jo, J., Kim, C. H. & Lee, J. H. Role of Si doping in reducing coercive fields for ferroelectric switching in HfO₂. *Phys. Rev. Appl.* **14**, 064012 (2020).
- 6 Park, M. H., Lee, Y. H., Kim, H. J., Kim, Y. J., Moon, T., Kim, K. D., Muller, J., Kersch, A., Schroeder, U., Mikolajick, T. & Hwang, C. S. Ferroelectricity and antiferroelectricity of doped thin HfO₂-based films *Adv. Mater.* **27**, 1811-1831 (2015).
- 7 Ghosh, G., Kang, Y., King, S. W. & Orlowski, M. Role of CMOS back-end metals as active electrodes for resistive switching in ReRAM cells. *ECS J. Solid State Sci. Technol* **6**, N1-N9 (2017).
- 8 Ma, C., Luo, Z., Huang, W. C., Zhao, L. T., Chen, Q. L., Lin, Y., Liu, X., Chen, Z. W., Liu, C. C., Sun, H. Y., Jin, X., Yin, Y. W. & Li, X. G. Sub-nanosecond memristor based on ferroelectric tunnel junction. *Nat. commun.* **11**, 1439 (2020).
- 9 Abuwasib, M., Serrao, C. R., Stan, L., Salahuddin, S. & Bakaul, S. R. Tunneling electroresistance effects in epitaxial complex oxides on silicon. *Appl. Phys. Lett.* **116**, 032902 (2020).
- 10 Bakaul, S. R., Serro, C. R., Lee, M., Yeung, C. W., Sarker, A., Hsu, S. L., Yadav, A. K., Dedon, L., You, L., Khan, A. I., Clarkson, J. D., Hu, C. M., Ramesh, R. & Salahuddin, S. Single crystal functional oxides on silicon. *Nat. Commun.* **7**, 10547 (2016).
- 11 Pozar, D. M. *Microwave Engineering (4th edition)*. John Wiley & sons, (2012).
- 12 Torrezan, A. C., Strachan, J. P., Medeiros-Ribeiro, G. & Williams, R. S. Sub-nanosecond switching of a tantalum oxide memristor. *Nanotechnology* **22**, 485203 (2011).
- 13 Abuwasib, M., Lee, H., Lee, J. W., Eom, C. B., Gruverman, A. & Singiseti, U.

- Characterization and modeling of Co/BaTiO₃/SrRuO₃ ferroelectric tunnel junction memory by capacitance–voltage (C–V), current–voltage (I–V), and high frequency measurements. *IEEE Trans. Electron. Devices* **65**, 2186-2191 (2019).
- 14 Wang, C., Wu, H. Q., Gao, B., Wu, W., Dai, L. J., Li, X. Y. & Qian, H. Ultrafast RESET analysis of HfO_x-Based RRAM by sub-nanosecond pulses. *Adv. Electron. Mater.* **3**, 1700263 (2017).
 - 15 Wang, C. C., Hou, Z. Y. & Deng, Y. L. 2-GHz 2×VDD 28-nm CMOS digital output buffer with slew rate auto-adjustment against process and voltage variations. *J. Circuits, Syst. Comput.* **29**, 2050088 (2019).
 - 16 Mielke, N. R., Frickey, R. E., Kalastirsky, I., Quan, M. Y., Ustinov, D. & Vasudevan, V. J. Reliability of solid-state drives based on nand flash memory. *Proc. IEEE* **105**, 1725-1750 (2017).
 - 17 Kim, S. W., Yang, J. H., Park, E. J., Choi, J. M. & Kwon, K. W. A high efficiency variable stage and frequency charge pump for wide range ISPP. *IEEE International Symposium on Circuits and Systems (ISCAS)*, 1-5 (2020).
 - 18 He, K. M., Zhang, X. Y., Ren, S. Q. & Sun, J. Deep residual learning for image recognition. *Proceedings of the IEEE conference on computer vision and pattern recognition (CVPR)*, 770-778 (2016).
 - 19 Xiao, H., Rasul, K. & Vollgraf, R. Fashion-MNIST: a novel image dataset for benchmarking machine learning algorithms. Preprint at <https://arxiv.org/abs/1708.07747> (2017).

Responses to Reviewer #3

Thank you very much for your pertinent comments and suggestions on our manuscript. We have carefully revised the manuscript and our responses to your comments are listed as follows:

Comment 1. *The authors refer to previous works stating that reducing the ferroelectric barrier thickness generally results in polydomains. Was the same phenomenon observed and verified with PZT for this work? And was the thickness of the ferroelectric thin film optimized this way?*

Answer 1: Thank you for raising this important topic. Following your suggestion, the evolution of the polydomains with reducing thickness (d) was verified for PZT by PFM, as shown in Fig. RIII-1. Consistent with the earlier reports¹, the domain size decreases with decreasing ferroelectric film thickness, which is beneficial for achieving multilevel resistances in FTJ devices. Interestingly, as shown in Fig. RIII-1b and c, the domain width (W) as a function of PZT thickness can be fitted by a power law:

$$W=Ad^{\gamma} \quad (\text{RIII-1})$$

with a scaling exponent $\gamma = 0.54$ and 0.53 for the out-of-plane and in-plane domains, respectively. This follows the famous Landau-Lifshitz-Kittel scaling law¹ with $\gamma = 0.5$.

To optimize the thickness of ferroelectric film, two aspects have been considered: 1) Reducing the thickness of the ferroelectric barrier is conducive to forming ferroelectric polydomains (see Fig. RIII-1), which is beneficial for achieving more conductance states. 2) The thickness of the ferroelectric thin film also influences the operation voltage. As shown in Fig. RIII-2, the resistance vs. pulsed voltage hysteresis loops for the FTJs with different PZT thicknesses were measured. With decreasing PZT thickness, the coercive voltages decrease as a result, which is beneficial for reducing operation voltages.

Besides, it is noted that with decreasing PZT film thickness, the FTJ resistance decreases as a result. As shown in Fig. RIII-2e and f, the extracted resistances for both ON and OFF states from Fig. III-2a and b clearly show exponential dependences on d , indicating the tunneling effect in FTJs².

Fig. RIII-1 is added as **Fig. 2**. **Fig. RIII-2** is added as **Supplementary Fig. S3**. The relevant descriptions have been added in **Lines 142-153 of page 8**, **Lines 228-235 of Page 12** of the revised manuscript and **S4** of the revised Supplementary information.

Fig. RIII-1 | Ferroelectric domain structures for (111)-oriented PZT films with different film thicknesses. a PFM out-of-plane and in-plane phases and topographies of PZT films with different thicknesses. Domain width *versus* film thickness of **b** out-of-plane and **c** in-plane domains.

Fig. RIII-2 | Ferroelectric film thickness dependence of coercive voltage and resistance for FTJs. **a, b** Resistance vs. pulsed voltage with pulse durations of $t_d = 10$ ns and $t_d = 630$ ps. **c, d** Thickness dependent coercive voltage with $t_d = 10$ ns and $t_d = 630$ ps. The solid lines are the fitting results by a power law $V_c \propto d^\alpha$. **e, f** Thickness dependent resistances for ON and OFF states with $t_d = 10$ ns and $t_d = 630$ ps. The solid lines are the fitting results by an exponential law.

Comment 2. *What was the motivation behind using a highly diffusive metal like silver as one of the electrodes? How to prevent the migration of Ag to form a diffusive memristor like what reported in the literature (see Nature materials 16 (1), 101) Considering the ultrathin nature of the ferroelectric film would not a metal like Platinum with work function greater than silver serve the purpose better?*

Answer 2: Thank you for raising this important topic.

1) Motivation for using Ag electrode with low work function. To investigate the effect of different electrodes, the FTJs with Pt, Cu, and Ag electrodes were

prepared and their resistance vs. pulsed voltage loops are shown in Fig. RIII-3. The resistance switching characteristics of the FTJs with different electrodes are listed in Table RIII-1. It can be seen that with increasing work function of electrodes from Ag to Pt, the ON/OFF ratio increases, while coercive voltages increase obviously and the subnanosecond resistance switching can hardly be realized for Pt electrode under affordable voltages. Considering that the ON/OFF ratio of >100 for the FTJ with Ag electrode is sufficient to meet the target performance for an synaptic device³, Ag electrode was then chosen to reduce the operation voltage and enhancing the operation speed.

Fig. RIII-3 is added as **Supplementary Fig. S4**. **Table RIII -1** is added as **Table S2**. The relevant descriptions are added in **Line 235 of Page 12** to **Line 240 of Page 13** of the revised manuscript and **S5** of the revised Supplementary information.

Fig. RIII-3 | FTJs with different metal electrodes. Resistance vs. pulsed voltage loops with different top electrodes by applying **a** 100 ns, **b** 10 ns, and **c** 630 ps pulsed voltages.

Table RIII -1 Resistance switching characters of FTJs with different electrodes

Top electrode	Pt	Cu	Ag
Work function	5.65 eV	4.5 eV	4.26 eV
Coercive voltage ($t_d = 100$ ns)	3.6 V/-3 V	1.1 V/2 V	0.7 V /-1.4 V
Coercive voltage ($t_d = 10$ ns)	8.2 V/-6.7V	2.7 V/-5 V	1.3 V/-1.7 V
Coercive voltage ($t_d = 630$ ps)	-	11.5 V/-17 V	4.7 V/-5 V
ON/OFF ratio	3000	1000	500

2) Exclude Ag migration induced resistance switching. Our FTJ is very different from the Ag diffusive memristor in *Nature materials* 16 (1), 101 in which the dielectric layer was co-sputtered with Ag⁴. Thus, as the reported results in this literature,

there are Ag nanocrystals formed in the dielectric and would diffuse to form conducting filaments under electric field. While for our FTJ devices, the Ag electrodes were sputtered after the preparation of the PZT ferroelectric films. The Ag nanocrystals were avoided in the barrier layer.

The resistance switching mechanism of our FTJ is closely related to the ferroelectricity-affected band structure instead of Ag migration. There are some experimental evidences to exclude the occurrence of Ag migration.

(1) The resistance switching characteristics based on the tunneling across a ferroelectric barrier of FTJs are different from these of Ag migration. For Ag conducting filament based memristors, the current increases abruptly when Ag filament formed¹⁶. Thus, the I - V curve is typically linear at ON state, and the current would decrease with increasing temperature because the conduction mechanism follows a metallic behavior¹⁷. While for our FTJs, as shown in Fig. RIII-4, the I - V curves at ON state follow a thermally-assisted tunneling model, suggesting a quantum tunneling mechanism. In addition, as shown in Fig. RIII-2, both ON and OFF states of resistances show exponential dependences on the PZT thickness, also indicating a tunneling effect in FTJs².

Fig. RIII-4 | I - V curves and transport mechanism analyses of the Ag/PZT/NSTO FTJ. **a** I - V curves measured by sweeping the voltage from 0 to 2 V, then 2 V to -2 V, and finally back to 0 V. The arrows indicate the voltage sweeping direction. **b, c** $\ln I_F$ - V at various temperatures from 270 to 150 K at ON and OFF states, respectively. **d, e** $\ln [J_s \cosh (E_0/k_B T)/T]$ vs. $1/E_0$ plots for ON and OFF states, respectively. The red solid lines are linear fitting results. **f** Energy profiles of ON and OFF states of Ag/PZT/NSTO FTJs by considering the work function of Ag (~ 4.26 eV), and electron affinities of PZT (~ 3.5 eV) and NSTO (~ 4.0 eV)^{6,7}.

(2) As shown in Fig. RIII-3, the resistance switching behavior of the FTJ with Ag electrode is similar to these with Cu and Pt electrodes, supporting the same underlying resistance switching mechanism among different electrodes⁸. This is also an evidence to exclude the occurrence of Ag filaments in the samples.

(3) The resistive switching of the Ag/BTO/NSTO FTJ is closely correlated with a nucleation-limited-switching (NLS) model of the ferroelectric domain dynamics⁹, as demonstrated in Fig. RIII-5.

Fig. RIII-5 | Ferroelectric domain switching dynamics for the (111)-oriented PZT in FTJs. **a** Schematic illustration of the applied voltage pulse sequence. **b** Resistance measured at 0.05 V versus pulse duration. The kinks are denoted by arrows. **c** Relative area fraction of the ferroelectric up domain versus pulse duration. The solid curves are the results of fitting by the NLS model. **d** Evolution of the mean switching time (τ_{mean}) as a function of the inverse of electric field ($1/|E|$). The solid lines are the fitting results by Merz's law⁹.

All the above experimental results confirm that the resistance switching of the Ag/BTO/NSTO FTJ is caused by ferroelectric polarization switching rather than the conduction bridge based on Ag filaments.

The relevant descriptions are added in **Lines 241-246 of Page 13** of the revised manuscript and **S6** of the revised Supplementary information.

Comment 3. *What is the grain size of the ferroelectric film at 1.2nm thickness? With 100um wide silver electrode, one would expect to capture a few grain boundaries which could prove detrimental for the device endurance.*

Answer 3: Thank you for raising this important topic. As you pointed out, the grain boundary may reduce the device endurance, because oxygen vacancies could accumulate at grain boundaries, pin the ferroelectric domain and degrade the insulation of devices¹⁰. For our FTJ device, the ultrathin ~1.2 nm PZT ferroelectric film is epitaxially grown on the (111)-oriented NSTO single crystal substrate, and the orientation is consistent with the substrate. As shown in Fig. RIII-6, the uniform orientation of the PZT/NSTO sample is revealed by the electron back scattered diffraction (EBSD) inverse pole figure (IPF) maps of 500×500 μm², indicating its monocrystal nature. To further investigate the possible grain boundaries in PZT film, the HAADF-STEM images were measured at the cross-section of a PZT/NSTO sample. The STEM sample was fabricated by focused ion beam and its lateral size is ~1.5 μm. This 1.5 μm sample was detected from one side to the other by measuring local HAADF-STEM images, and 5 representative pictures are shown in Fig. RIII-7. No grain boundary was observed in the tested area. However, this cannot entirely exclude the possibility that there exist a few grain boundaries which will be detrimental for the device endurance. Improving sample qualities by reducing grain boundaries is always important.

Fig. RIII-6 | Scanning electron microscopy (SEM) image and electron backscattered diffraction (EBSD) maps of PZT/NSTO. a Scanning electron microscopy (SEM) image corresponding to the following EBSD maps. **b, c, and d** electron back scattered diffraction (EBSD) inverse pole figure (IPF) maps: IPF-X, IPF-Y, IPF-Z of NSTO substrate, respectively.

Fig. RIII-7 | Representative HAADF-STEM images of the PZT/NSTO heterostructure. 5 representative HAADF-STEM images of (111)-oriented PZT (1.2 nm)/NSTO.

Fig. RIII-6 is added as **Supplementary Fig. S9** and **Fig. RIII-7** is added as **Supplementary Fig. S10**. The relevant descriptions are added in **Lines 271-273 of Page 14** of the revised manuscript and **S10** of the revised Supplementary information.

Comment 4. *In the ferroelectric domain switching dynamics section, the authors write that the FTJ was first SET to ON state but at same time use V_RESET to label the voltage pulse. This appears confusing and contradicting. Usually, it is referred to as V_SET in memristive devices nomenclature. Since, the authors also refer to some of the works on memristive devices it would make more sense if they made this change.*

Answer 4: Thank you for pointing out this mistake. The relative labels were corrected, as shown in **Fig. 3** in the revised manuscript.

Comment 5. *Endurance of this device is exceptional with ON/OFF ratio degrading very little over 10⁹ cycles. It would be interesting to study its dependence on top electrode, for example Silver versus Platinum which can make the FJT more robust. It would be appreciated by the research community if results of such experiment were shared.*

Answer 5: We appreciate the reviewer's suggestion. Following your suggestion, the switching endurances of FTJs with Pt and Ag electrodes were measured using the same

pulse duration $t_d = 1 \mu\text{s}$, as shown in Fig. RIII-8. For the FTJ with Pt top electrode, larger operation voltages of 3.5 V/-4.5 V were used to realize the ON/OFF ratio ~ 1500 , and the corresponding switching endurance is $\sim 2 \times 10^6$. By decreasing applied voltages to 2.7 V/-3.2 V, the ON/OFF ratio decreases to ~ 300 , as shown in Fig. III-8b, and the corresponding switching endurance increases to $\sim 10^8$. While for FTJ with Ag electrode with an ON/OFF ratio of ~ 300 , the switching endurance is as high as 6×10^8 . This should be due to the lower operation voltages of 1.5 V/-2.5 V for the FTJ with Ag electrode.

Fig. RIII-8 | Switching endurance of FTJs with Pt and Ag top electrodes. a, b Endurance of Pt electrode FTJ switched by applying 3.5 V/-4.5 V and 2.7 V/-3.2 V pulsed voltages, respectively. **c** Endurance of Ag electrode FTJ by applying 1.5 V/-2.5 V voltage pulses. The pulse durations t_d in **a**, **b**, and **c** are 1 μs .

Fig. RIII-8 is added as **Supplementary Fig. S5**. The relevant descriptions are added in **Lines 240-241 of Page 13** of the revised manuscript and **S5** of the revised Supplementary information.

Comment 6. *For sub-nanosecond pulse measurement, usually a coplanar waveguide device structure is made to ensure the ultra-fast pulse maintains its shape when it is delivered to the device (see Advanced Functional Materials 26 (29), 5290-5296). How did the authors ensure their sub-ns measurements?*

Answer 6: Thank you for raising this important point. To carry out the ultrafast measurements, our FTJ was connected to the test circuit through a microstrip waveguide^{11,12}. The characteristic impedance $Z_0 = 50 \Omega$ of the microstrip waveguide matches with the connection cables (0-18 GHz, Mini-circuits, USA), pulse generator (PSPL10300B, Tektronix, USA), oscilloscope (DSA70804, Tektronix, USA) and switch matrix (RC-4SPDT-A18, 0-18 GHz, Mini-circuits, USA), and the connection wire between the FTJ and waveguide is as short as possible. To characterize the transmission capability, the microstrip waveguide was used to test a short copper wire (the same type of copper wire is also used to connect the FTJ to the waveguide) by

using a vector network analyzer (AV3656A, CETC 41, China), as the transmission and reflection properties up to 3 GHz shown in Fig. RIII-9a and b. It can be seen that the magnitude of transmission through the waveguide is high (≥ -2.2 dB) and the shift of transmitted phase varies linearly with frequency, indicating the weak dispersion which ensures the pulse integrity¹³. Importantly, the pulsed voltages before and after passing through the waveguide were measured by an oscilloscope, as shown in Fig. RIII-9c. The pulse ~ 600 ps shows little shape deformation after transmitted through the waveguide. These results suggest that the voltage pulse maintains its shape when it is delivered to the device.

Fig. RIII-9 | High frequency characterizations of microstrip waveguide. a Magnitudes and **b** phases of transmission and reflection. **c** The ~ 600 ps waveforms before and through the microstrip waveguide.

Fig. RIII-9 is added as **Supplementary Fig. S13**. The relevant descriptions are added in **Methods** of the revised manuscript and **S13** of the revised Supplementary information.

Comment 7. *High Temperature and PLD were used to obtained high quality of epitaxial growth of the ferroelectric film. Can the authors comment on the possibility of using more industry-friendly-tools, such as sputtering, to prepare such films? How would the FTJ performance of sputtering deposited films compare with the PLD deposited devices?*

Answer 7: Thank you for raising this interesting topic. Following your suggestion, the PZT-based FTJs were prepared by magnetron sputtering as well. The fabrication conditions are as follows: The PZT film was grown by magnetron sputtering at 480 °C under 10 mTorr flowing gas of Ar : O₂=27 : 3. After growth, the films were cooled to room temperature at a rate of 5 °C/min in 250 Torr O₂ atmosphere. Then, the PZT film was annealed at 650 °C in atmosphere for 1 hour. The rate of rising and decreasing temperature was 3 °C/min during annealing process. The resistance vs. pulsed voltage

loop of FTJs fabricated by PLD and sputtering are compared in Fig. RIII-10. The sputtering prepared FTJs can be operated at subnanosecond as well. However, the performances of sputtered FTJ, such as ON/OFF ratio, are poorer than these for the FTJ fabricated by PLD. This may be due to the composition deviation during sputtering¹⁴. We believe that through further optimizing the growth condition, the performance of sputtered FTJs can be improved.

Fig. RIII-10 | Resistance vs. pulsed voltage loops of FTJs grown by sputtering and PLD with $t_d = 630$ ps.

Comment 8. Please clearly list what properties were measured on the 100 μ m devices and 80nm devices, respectively.

Answer 8: We thank the reviewer for pointing out this issue. Only the **Supplementary Fig. S5** in previous manuscript was achieved from 80 nm diameter FTJ devices. All the other results in the manuscript were obtained from 100 μ m diameter FTJ devices. Recently, we succeeded in achieving 50 nm diameter FTJ, and the related results are used to replace previous results on 80 nm diameter FTJ devices, as shown in **Supplementary S8** of revised Supplementary information.

The relevant results for the sample with diameter of 50 nm was added to Supplementary S8 to clearly clarify this information.

References

- 1 Gatalan, G., Béa, H., Fusil, S., Bibes, M., Paruch, P., Barthélémy, A. & Scott, J. F. Fractal dimension and size scaling of domains in thin films of multiferroic BiFeO₃. *Phys. Rev. Lett.* **100**, 027602, (2008).
- 2 Li, C. J., Huang, L. S., Li, T., Lv, W. M., Qiu, X. P., Huang, Z., Liu, Z. Q., Zeng, S.

- W., Guo, R., Zhao, Y. L., Zeng, K. Y., Coey, M., Chen, J. S., Ariando & Venkatesan, T. Ultrathin BaTiO₃- Based ferroelectric tunnel junctions through interface engineering. *Nano Lett.* **15**, 2568-2573 (2015).
- 3 Xi, Y., Gao, B., Tang, J. S., Chen, A., Chang, M. F., Hu, X. B. S., Van Der Spiegel, J., Qian, H. & Wu, H. Q. In-memory learning with analog resistive switching memory: a review and perspective. *Proc. IEEE* **109**, 14-42 (2020).
 - 4 Wang, Z. R., Joshi, S., Savel'ev, S. E., Jiang, H., Midya, R., Lin, P., Hu, M., Ge, N., Strachan, J. P., Li, Z. Y., Wu, Q., Barnell, M., Li, G. L., Xin H. L., Williams, R. S., Xia, Q. F. & Yang, J. J. Memristors with diffusive dynamics as synaptic emulators for neuromorphic computing. *Nat. Mater.* **16**, 101-108 (2016).
 - 5 Yoon, J. H., Wang, Z. R., Kim, K. M., Wu, H. Q., Ravichandran, V., Xia, Q. F., Hwang, C. S. & Yang, J. J. An artificial nociceptor based on a diffusive memristor. *Nat. Commun.* **9**, 417 (2018).
 - 6 Scott, J. F., Watanabe, K., Hartmann, A. J. & Lamb, R. N. Device models for PZT/Pt, BST/Pt, SBT/Pt, and SBT/Bi ferroelectric memories. *Ferroelectrics* **225**, 83-90 (1999).
 - 7 Guo, D. Y., Liu, H., Li, P. G., Wu, Z. P., Wang, S. L., Cui, C., Li, C. R. & Tang, W. H. Zero-power-consumption solar-blind photodetector based on β -Ga₂O₃/NSTO heterojunction. *ACS Appl. Mater. Interfaces* **9**, 1619-1628 (2017).
 - 8 Hernandez-Martin, D., Gallego, F., Tornos, J., Rouco, V., Beltran, J. I., Munuera, C., Sanchez-Manzano, D., Cabero, M., Cuellar, F., Arias, D., Sanchez-Santolino, G., Mompean, F. J., Garcia-Hernandez, M., Rivera-Calzada, A., Pennycook, S. J., Varela, M., Muñoz, M. C., Sefrioui, Z., Leon, C. & Santamaria, J. Controlled sign reversal of electroresistance in oxide tunnel junctions by electrochemical-ferroelectric coupling. *Phys. Rev. Lett.* **125**, 266802 (2020).
 - 9 Boyn, S., Grollier, J., Lecerf, G., Xu, B., Locatelli, N., Fusil, S., Girod, S., Carrétéro, C., Garcia, K., Xavier, S., Tomas, J., Bellaiche, L., Bibes, M., Barthélémy, A., Saïghi, S. & Garcia, V. Learning through ferroelectric domain dynamics in solid-state synapses. *Nat. Commun.* **8**, 14736 (2017).
 - 10 Park, M. H., Lee, Y. H., Mikolajick, T., Schroeder, U. & Hwang, C. S. Review and perspective on ferroelectric HfO₂-based thin films for memory applications. *MRS Commun.* **8**, 795-808 (2018).
 - 11 Pozar, D. M. *Microwave Engineering (4th edition)*. John Wiley & sons, (2012).
 - 12 Choi, B. J., Torrezan, A. C., Strachan, J. P., Kotula, P. G., Lohn, A. J., Marinella,

- M. J., Li, Z. Y., Williams, R. S. & Yang, J. J. High-speed and low-energy nitride memristors. *Adv. Funct. Mater.* **26**, 5290-5296 (2016).
- 13 Torrezan, A. C., Strachan, J. P., Medeiros-Ribeiro, G. & Williams, R. S. Sub-nanosecond switching of a tantalum oxide memristor. *Nanotechnology* **22**, 485203 (2011).
- 14 Lin, C., Sun, D. C., Ming, S. L., Jiang, E. Y. & Liu, Y. G. Magnetron facing target sputtering system for fabricating single-crystal films. *Thin Solid Films* **279**, 49-52 (1996).

REVIEWERS' COMMENTS

Reviewer #1 (Remarks to the Author):

The authors have addressed the reviewer's comments satisfyingly and the manuscript can be published in Nature Communications as it is.

Reviewer #2 (Remarks to the Author):

The authors have carried out additional experiments and studies, with supplementary data shown in the revised manuscript. The responses and revisions made are acceptable. I suggest acceptance of the article.

Reviewer #3 (Remarks to the Author):

The authors have addressed my previous questions in the revision, which might be accepted now.